# Impact of RMB internationalization on China's competitiveness in financial services trade based on the VAR model: Evidence from China-US

**Yufei Lei** ©*

Department of Finance, School of Finance, Zhongnan University of Economics and Law, Wuhan, Hubei, China

* lyflyf960927@163.com

**Data Availability Statement:** All relevant data are within the manuscript and its Supporting Information files.

## Abstract

An increase in a currency internationalization levels can positively impact its credibility in international economic activities, and expand the effective demand and optimize the supply structure for the country's financial service trade. In this way, a state can improve its financial service trade competitiveness in the international market. This study builds a vector autoregressive model based on time-series data of China-US financial services trade from 2010 to 2021, analyzes the impact of different quantitative indicators of RMB internationalization on this trade from the impulse response results, and validates the conclusions using various inspection methods. The results show that the increase in RMB internationalization helps to narrow the China-US financial services trade balance, but with a significant lag. And this effect is heterogeneous in different dimensions, demonstrated by the fact that the development of overseas RMB securities business is more important for the level of RMB internationalization to narrow the China-US financial services trade balance. Finally, among the specific measures to improve its financial services trade, China should focus on developing the international competitiveness of the traditional RMB deposit and loan financial sector, while the competition in the overseas market for high value-added financial businesses must also not be neglected. Furthermore, China needs to implement more targeted RMB internationalization development policies at different levels in the future to provide high-quality financial services to the rest of the world and aid in the economic recovery of the world in the "post-pandemic" era.

## 1. Introduction

The financial services industry has become the "blood" of a country's national economy, vital to achieve modernization and development [1]. In an open modern economy, financial services trade formed by cross-border transactions in financial services has not only become an important part of the financial industry and service trade, but also an important channel for

**Funding:** The author(s) received no specific funding for this work.

**Competing interests:** The authors have declared that no competing interests exist.

the flow and allocation of domestic and foreign factors. The development level of a country's financial service trade not only reflects the quality of its domestic financial industry development but also the connection between its domestic and international circulation to a certain extent.

Worldwide, the financial industry in developed countries have emerged early in the country's development history and developed rapidly thereafter. The sound industrial system and legal framework in these countries have enabled the industry to maintain healthy and stable development. In particular, the financial services industry of the US, the global financial center, is prestigious worldwide. The US dollar has long been recognized as the world's global currency, and played a vital role in enabling the US to occupy a key position in the world's financial field. By contrast, China's foreign financial services trade has been in deficit for a long time, and its international competitiveness is relatively weak [2]. According to the 2021 data from the World Trade Organization and the United Nations Conference on Trade and Development (UNCTAD) data center, US financial service trade export volume was USD 171.74 billion, the import value was USD 49.529 billion, and international market share was 26.83%. In contrast, the export volume of China's financial services trade was only USD 5.107 billion, the import volume was USD 5.346 billion, and international market share was only 1.26%. Moreover, the bilateral financial service trade volume between China and the US has been in deficit all year round. Still although the current international market share of China's financial services trade is low, its average annual growth rate is relatively strong, and its import and export structure is relatively balanced. This demonstrates shows its huge development potential. Many factors affect the competitiveness of financial services trade, coupled with the increase in unstable and uncertain factors in the current international situation. Therefore, an in-depth and detailed research of various factors that affect the competitiveness of financial services trade is needed. Based on this, China should take active measures to rapidly improve the competitiveness of its financial services trade and accelerate its economic recovery.

In recent years, China's foreign trade development has faced major challenges. As a payment medium for international trade, currency internationalization plays an important role in promoting trade development and reducing trade uncertainty. Then, one may ask: Is a country's currency internationalization level and its financial services trade competitiveness significantly correlated? Since the outbreak of the global financial crisis in 2008, the drawbacks of the international monetary system based on US dollar standard have become increasingly apparent. Countries worldwide have sought to reform the international monetary systems. Simultaneously, China began to actively promote the internationalization of the RMB, including gradually relaxing the scope of RMB cross-border trade settlements, actively signing bilateral currency swap agreements with other economies worldwide, and establishing an RMB cross-border payment system. For instance, in 2009, China signed local currency swap agreements with South Korea, Hong Kong and other countries or regions. According to the People's Bank of China's(PBOC) "RMB Internationalization Report 2021," as of December 2020, the amount of RMB swaps has reached 3.99 trillion yuan, and the cumulative contracted amount is 23.8188 trillion yuan. The PBOC has also set up RMB clearing banks in countries and regions such as Hong Kong, Germany, and the United Kingdom to promote the matching of financial infrastructure. On November 30, 2015, the International Monetary Fund (IMF) announced that the RMB would be officially included in the IMF's Special Drawing Rights currency basket. The directly promoted the demand for RMB in the international market. According to IMF and Society for Worldwide Interbank Financial Telecommunications(SWIFT) data, the RMB accounted for 2.79% of international foreign exchange reserves in the fourth quarter of 2021, and the RMB's international payment share was 3.2% in December 2021. With ever increasing shares in global foreign exchange reserves and global payments reaches, the RMB is

playing an increasingly important role in the international financial market [3]. Although the RMB's current currency status is not high, China is pushing it toward internationalization [4]. China's two main goals are to promote the use of RMB in foreign trade by increasing its attractiveness and availability, and encourage its use as a store of value in international finance. In a general sense, the internationalization of the RMB is the process by which it crosses national borders circulates overseas and becomes a world-recognized valuation, settlement and reserve currency. As an important means for China to actively integrate into the world currency system, the internationalization of the RMB provides a favorable opportunity for the rapid development of China's financial service trade and narrowing the industry gap with developed countries such as the US and Europe.

In summary, from a macroeconomic perspective, multiple factors affect the impact of different government policies on the economy [5]. This study empirically analyzes the impact of RMB internationalization on China-US financial services trade and the influence of various indicators on this trade, and determines the shortcomings in China's financial services trade. The aim is shed light on new ways through which the Chinese government can enhance the competitiveness of financial services trade. Data from UNCTAD, the National Bureau of Statistics of China, and the PBOC from 2010 to 2021 are analyzed using a vector autoregressive model, and the methods of Sims [6] and Wang [7] to establish VAR model. Finally, measures for promoting the development of China's financial services trade are suggested. We find that RMB internationalization can improve the level of China's financial services trade, but there is an obvious lag. Specifically, RMB internationalization promotes China's financial services trade by increasing the holding of RMB financial assets by overseas institutions and individuals. However, the degree of influence is heterogeneous in different fields. Thus, China's financial authorities need to take specific measures in different directions for the future development of the financial services industry.

This study's contributions are three-fold. First, this study is among the few which shed light on the influencing factors of financial services trade competitiveness from the perspective of RMB internationalization. Second, in terms of research objects, extant research on the impact of RMB internationalization on trade mainly focuses on trade in goods and services. As a currency, the degree of internationalization of the RMB directly affects financial service areas that are more affected by monetary functions. This study expands the analytical perspective of RMB internationalization to China-US financial service trade, enriching research on how the transmission effect of RMB internationalization influences China's economic development. Third, in terms of variables that reflect the level of RMB internationalization, this study focuses on selecting indicators that can better reflect the actual application value of the RMB(settlement, foreign direct investment (FDI), and financial business), rather than focusing too much on the measurement of financial trade competitiveness or quantifying the level of RMB internationalization from an overly macroscopic perspective, as in previous studies. This new perspective on measuring the internationalization of the RMB can provide more targeted policy suggestions for China in choosing a path to improve the competitiveness of financial services trade.

## 2. Literature review

The extensive literature on China's financial services trade and RMB internationalization can be summarized into the following three aspects.

The first aspect concerns the measurement of the RMB internationalization level. Research on this topic mainly comes from China and generally involves econometric analyses based on the international currency function and the influencing factors of RMB internationalization.

Earlier studies were relatively simple and direct. For example, Li [8] performed a simple arithmetic average on the proportion of currency international reserves, scope of currency circulation, and total size of currency circulation abroad, and used it to measure the level of currency internationalization; Li and Liu [9] started with the three major indicators-international reserves, trade settlement, and international bonds-and used the proportion of currency functions of various countries as the framework to predict the future development trend of RMB internationalization.

Among recent studies, Tung et al. [10] focused on the three major functions of currency and used seven major share indicators to calculate the currency internationalization index: the proportions of countries whose exchange rate is pegged to the currency, trade pricing, foreign monetary assets and liabilities of the issuing country's bank, foreign exchange transactions, international reserves, and denominated interim bonds. They then used principal component analysis to determine the weight of each indicator, and then measured the internationalization index of several major currencies in circulation worldwide. Orăştean [11] calculated the internationalization level of the corresponding currency based on the proportion of the gross domestic product (GDP) of 30 countries in the world and the proportion of foreign exchange market transactions in the country. The author found that the internationalization level of the RMB has greatly improved compared with the past, but is still at a low level compared with major international currencies. Liu et al. [12] arrived at the same conclusion using the proportion of RMB cross-border trade settlements, financial market transactions, and reserve currency to measure the degree of currency internationalization. Yu and Wang [13] empirically found that RMB internationalization is affected by current account surplus and RMB appreciation. In contrast, the capital account inhibits RMB internationalization. Based on the three major functions of currency, Bénassy-Quéré [14] divided the variables affecting currency internationalization into fundamental and structural factors, and constructed a total index of currency internationalization level using an ordinary least squares regression and principal component analysis. Furthermore, the author split the total index into the absolute and relative indices. The author found that the internationalization level of the RMB is far lower than that of the US dollar and euro, but the development trend has great potential. Fan et al. [15] used the entropy weight method to determine the weight of the global aggregate indicators of each function of currency internationalization and then assigned a certain weight to the share indicators corresponding to each major currency, thereby calculating the internationalization index of several major currencies in circulation worldwide.

The second aspect studies focus on is the economic effect of RMB internationalization. As an emerging economy, China needs to be prepared at all times to deal with the possible impact of policy changes in developed Western economies and changes in the international economic situation [16]. Therefore, in recent years, analysis in this field has mainly focused on the macro level. Examining the balance of RMB deposits in Hong Kong, Ma [17] found that RMB internationalization can buffer the adverse impact of RMB exchange rate appreciation on exports. Yu and Wang [18] focused on the currency swap agreement between China and other countries, which is a specific RMB internationalization measure, to explore the impact of RMB internationalization on exports and FDI. The authors found that the exchange agreement can buffer exchange rate risk, and promote a subsequent increase in China's exports and FDI. The research of Joseph and Kent [19] showed a complementary relationship between RMB internationalization, and import and export trade, which not only promote but also check and balance each other. He [20] pointed out that an improvement in currency internationalization promotes the development of the country's foreign trade. Based on the national-level trade data, Pu et al. [21] found that RMB internationalization can promote the growth of China's export scale.

Ma and Cao [22] studied the long-term relationship between RMB internationalization and China's trade competitiveness based on the VAR model. The authors showed that the expansion of the RMB settlement scale and the development of offshore markets help enhance China's trade competition. Deng and Peng [23] used the dynamic stochastic general equilibrium model to calibrate with real data and found that promoting the internationalization of the RMB can reduce the domestic output gap, inflation rate, and exchange rate fluctuations, thereby promoting the improvement of social welfare. Lu [24] used construction of the VAR-BEKK-Generalized AutoRegressive Conditional Heteroskedasticity(GARCH) model that, in the short term, the internationalization of the RMB significantly impacts the RMB exchange rate and interest rate, and an increase in the degree of internationalization increases the appreciation of the RMB and market interest rates. Meanwhile, monetary policy positively affects RMB internationalization through the exchange rate, but it is difficult for RMB internationalization to influence the effectiveness of monetary policy.

The third aspect research focuses on are the factors affecting the competitiveness of the financial services trade. Scholars have mainly combined traditional trade theory with practical cases. Trade in financial services, like trade in goods, reflects a certain level of sophisticated knowledge and skilled labor. First, foreign studies on the factors affecting trade in financial services are mostly concentrated within the framework of classical international trade theory. These theories hold that the root cause of trade advantages in financial services can be determined by studying the differences in human capital, technological factors, and physical capital. Ter [25] (1995) studied the comparative advantages of the banking industry and found that economies of scale is the main reason why the international banking industry provides cross-border services trade in the form of commercial existence, such as representative offices, branches, and subsidiaries. Sapir and Lutz [26] argued that there is no difference in the applicability of the comparative advantage theory between services trade and traditional commodity trade. Based on a series of empirical models, the authors argued that if a country has abundant human capital, then the services industry (such as the insurance sector) with relatively intensive human capital in the services trade has a comparative advantage. Conversely, if a country has abundant physical capital, then the services industry (such as the transportation sector) that has a large demand for physical capital in the country's services trade has a comparative advantage. Moshirian [27, 28] contended that financial services trade is similar to manufacturing and that some natural advantages of developed countries (including strong physical and human capital, large international assets of banks, and relatively transparent financial information disclosure) provide them with a comparative advantage in international financial services.

Li et al. [29] incorporated financial services trade into the analytical framework of intra-industry trade theory, and used it to explain the emergence, development, and internal mechanism of financial services trade. Studies have argued that international trade is mainly explained by inter-industry trade, and its theoretical premise is perfect competition; that is, compared with a country that formed a professional production advantage in a certain product or service and exported it, the importing country is relatively disadvantaged in the product or service. However, Li et al. argued that trade in financial services is more applicable to the theory of intra-industry trade; that is, when a country produces and exports a certain product or service in a certain industry, it must import the same product or service. Helpman and Krugman [30] and Markusen and Venables [31] argued that FDI is an important factor leading to inter-industry trade. Specifically, both physical and human capital play important roles in the financial services industry. Therefore, the main factors affecting the intra-industry trade of the financial services industry are the FDI scale in financial services, the level of the demand structure among trading countries (differences in the per capita income of trading partner countries), and factor endowments (differences in the amount of human capital in the financial

services industry of trading partner countries), and so on. According to Khoury and Sawides [32], the economic effect of trade on financial services is closely related to the development level of a country's services industry. The higher the level of the services industry, the higher the economic effect of trade liberalization in financial services.

Second, an increased number of domestic studies explore the factors affecting China's financial services trade. Dou [33] found that the competitiveness of China's financial services trade is affected by many factors, such as factor conditions (capital, manpower, technology, etc.), demand conditions, related (supporting) industries, enterprise organization, strategy and competition, and government role and opportunities,. Liu [34] showed a long-term equilibrium relationship between China's financial services trade competitiveness and GDP, foreign investment inflows, and money supply, while the short-term impact has a lag.

In addition, as the degree of opening up of the Chinese financial market increases, its impact on the competitiveness of China's financial services trade is receiving increasing attention. Huang and Deng [35] found that China's financial services trade exports are competitive in countries along the "Belt and Road." In the future, financial services exports to these countries should be strengthened to enhance the international competitiveness of financial services exports. Lyu et al. [36] found that financial liberalization can significantly enhance the international competitiveness of financial services trade by expanding its export scale of financial service trade and mitigating the impact of trade frictions on bilateral trade.

Notably, research on the measurement of the level of RMB internationalization is mostly analyzed at the macro and national levels, with few studies paying attention to reflecting residents' daily financial business on the level of RMB internationalization. In addition, there is insufficient attention on the role of RMB internationalization in research on the influencing factors of China's financial services trade. Further, research on the economic effect of RMB internationalization mainly focuses on commodity trade and monetary policy.

Considering these researches, this study examines the economic effects of RMB currency internationalization from the perspective of the China-US financial services trade. Referring to the measurement methods of RMB internationalization in the literature, this study constructs an evaluation index system for the level of RMB internationalization from the perspective of the RMB serving as a transaction medium and the financial business of economic individuals. Compared with the macro perspective in prior works, this study's focus is closer to the daily practical application of financial business entities. In addition, this study establishes a VAR to empirically examine how RMB internationalization affects China-US trade in financial services, and validates whether RMB internationalization can promote the reduction in the balance in financial services trade between China and developed countries. Finally, based on the empirical results, this study provides policy inspirations for China to strengthen the internationalization of the RMB and enhance the competitiveness of its financial services trade.

## 3. Research methods

### 3.1 Theoretical analysis and research hypothesis

As a national sovereign currency, the RMB functions as a medium of exchange, unit of account, and store of value at the official and private levels. RMB internationalization can extend these domestic functions of the currency to the international level [37]. Thus, it is beneficial to eliminate the obstacles restricting the development of financial services trade at both the supply and demand levels. Accordingly, this study argues that the improvement in a country's currency internationalization level can not only directly reduce the explicit transaction costs of its financial services trade, but also reduce the hidden costs brought about by exchange rate fluctuations, thereby expanding the effective demand for financial services trade. Besides,

the improvement in the currency internationalization level will also drive the development of cross-border financial services businesses and the increase in the types of financial service products, which improve the quality of financial services supply. The detailed explanation is as follows:

First, the transaction costs are reduced. In the financial services trade, the differences in currency types between the two parties often result in additional transaction costs. These costs include not only the currency exchange cost of financial services but also the procedural and time costs related to currency exchange. Currency internationalization reduces transaction procedures in financial services trade, saves exchange and time costs, and thus, increases the willingness of both the supply and demand sides involved in the financial services trade to undertake the transaction. For instance, consider overseas consumption. When consumers purchase financial services abroad, they first need to convert their local currency into US dollars for transactions. If the financial services provider does not accept US dollars, the consumer needs to convert the US dollars into the currency of the supplier's country again; that is, two instances of currency exchange are performed in one transaction, which greatly increases the time cost of the financial services trade and reduces consumers' willingness to trade. Currency internationalization increases the recognition and desirability of the local currency in other countries and regions, expands its scope and areas of use, reduces the currency exchange procedures in cross-border transactions of financial services, and creates more convenient conditions for financial services trade. Regarding the RMB, the "2021 RMB Internationalization Report" issued by the PBOC pointed out that in recent years, the RMB has been widely recognized in neighboring countries and countries along the "Belt and Road," and its usage has continued to increase. In 2019, the amount of cross-border RMB settlement between China and neighboring countries was approximately 3.6 trillion yuan, a year-on-year increase of 18.5%. In the same year, the RMB had direct transactions with the Singapore dollar, Malaysian ringgit, Thai baht, etc., greatly reducing the transaction costs of cross-border financial services trade.

Second, RMB internationalization can mitigate the risk of exchange rate fluctuations. Dai and Zhang [38] found that the RMB real effective exchange rate fluctuations have a much greater negative impact on China's exports of new services trade (including financial and insurance services) than on traditional services trade. Currency exchange in the during financial services trade is closely related to the exchange rate, and the exchange rate market usually fluctuates frequently and has high risks. The exchange rate fluctuation risk has a greater impact on financial services trade than on traditional services trade. When consumers purchase financial services abroad, the depreciation of the local currency brings additional losses to consumers when the foreign currency is priced at a certain level. The internationalization of a currency improves its international recognition so that direct delivery can help avoid the risk of exchange rate fluctuations in the foreign exchange market and reduce the uncertainty of cross-border transactions of financial services. Thus, improving the international influence of currency can not only directly reduce the explicit transaction cost of financial services trade but also reduce the implicit transaction cost by avoiding exchange rate fluctuations, thereby enhancing the competitiveness of financial services trade.

Third, the quality of financial service supply will improve. An increase in a country's currency internationalization level is often accompanied by an improvement in its financial industry development level and openness. The increase in demand for financial services from overseas consumers can also drive the development of the domestic financial industry, which can not only increase the types and scale of financial services products supplied but also help improve the quality of financial services. In addition, currency internationalization is conducive to the development of financial services providers abroad, and improves the international competitiveness of financial service trade by providing diversified and high-quality financial

services to consumers in the host country. For example, when the domestic currency has a high level of internationalization, the branches of banks and other financial institutions established abroad provide financial services to local consumers without the need for a third-party intermediary currency, thereby promoting the diversification of the cross-border supply of financial services and the high-quality development of trade in financial services.

However, most recent studies suggest that the dollar-dominated international monetary system will remain unshakable in the foreseeable future. Zheng [39] pointed out that the international monetary system dominated by the US dollar has existed for more than 70 years, which has created the inertia of demand for the US dollar; this demand is mainly reflected in trade, investment and hedging demands. Ranaldo [40] also pointed out that the benefits of a single sovereign currency system, such as the US dollar, lie in the following points: avoiding the moral crisis of international currency issuance, solving incentive incompatibility and the Triffin dilemma, eliminating the wealth effect of international currency, and restoring public product attributes. In addition, Eichengreen [41] and Cooper [42] believed that China's financial market is underdeveloped due to the lack of a good capital market, especially the national bond market, and the existence of relatively serious capital account controls; this means that the RMB may take several decades to become an international currency. These authors predicted that after the financial crisis, the world would gradually move towards a new international monetary system in which the US dollar will form a diversified international currency together with the euro, RMB and Special Drawing Rights, but this is not likely to happen in the short to medium term. Therefore, the improvement in the level of RMB internationalization has no obvious effect on narrowing the gap in financial services trade between China and the US in the short term, while it may have a strong promotion effect in the medium and long-term time. Still, based on the above analysis, this study proposes hypothesis 1:

**Hypothesis 1:** *An increase in the level of RMB internationalization narrows the gap in financial services trade between China and the US, but with a lag.*

As mentioned above, the role of RMB internationalization in promoting the competitiveness of financial services trade is mainly achieved by increasing the recognition and scope of use of the RMB in international financial service transactions. Therefore, the promotion effect on the competitiveness of financial services trade appears only when the level of RMB internationalization continues to increases significantly. Rather, in real international economic activities, one may ask how should RMB internationalization be quantified or which variables can reflect the level of RMB internationalization.

Some relatively mature indicators exist for measuring the degree of RMB internationalization. The RMB Internationalization Index (RII), compiled and released by the International Monetary Institute of Renmin University of China, is an existing and widely recognized indicator for measuring the degree of RMB internationalization. The RII is a comprehensive quantitative index aimed at objectively describing the actual use of the RMB in international economic activities. Cui [43] argued that this indicator can not only track the development dynamics of the RMB share in the three aspects of global trade valuation, financial transactions, and foreign exchange reserves, but also makes it easier to make horizontal comparisons with other major international currencies. However, other studies note some drawbacks. With international and domestic monetary system reforms, a specific index derived from a relatively fixed underlying logic and calculation framework may gradually have limitations over time. Even if scholars are constantly improving it, the lag caused by various human factors will lead to deviations, making it impossible to quantify RMB internationalization in a timely and comprehensive manner. Therefore, this study argues for more objective indicators to measure RMB internationalization.

Specifically, first, although the RII has disadvantages that cannot be ignored, there are still merits in constructing the index from the perspective of the currency function. Hao [44] believed that becoming a widely recognized medium of exchange in the world market and a world currency is the main development goal of the current stage of RMB internationalization. Sha et al. [45] measured the level of RMB internationalization based on the principal component analysis method and found that the improvement in RMB internationalization was mainly driven by acting as a trade medium. Generally, a limitation of this traditional measurement method is that it does not consider the market influence of a country's currency on other countries. Most methods measure the international influence of a currency at the all-sector level, ignoring the heterogeneity of the currency's influence in different industries. Therefore, this study considers the selection of variables in the RII for the function of the currency transaction medium. Specifically, the RMB ratios of international trade settlement and global direct investment are selected as the endogenous variables, as they reflect the current level of RMB internationalization at a relatively macro level.

Secondly, Ding [46] argued that RMB internationalization can be defined from the perspective of stakeholders involved its internationalization. Specifically, the benefits of RMB internationalization mainly come from cross-border holdings. From a dynamic perspective, the currency internationalization of emerging economies can be expressed as "non-residents are able and willing to use or hold the country's currency." "Able and willing" reflects the relationship between supply and demand in the process of currency internationalization; "use or hold" reflects the two levels of currency internationalization, including cross-border use for pricing and settlement in international trade and financial fields, and cross-border holdings of the country's currency and currency-denominated financial products by non-residents. Moreover, cross-border use of and holdings in RMB are mutually reinforcing. Cross-border use is the foundation for cross-border holdings, which in turn can deepen cross-border use and expand the scale of cross-border settlements. As China's financial market liberalizes, the link between Chinese financial and international assets becomes stronger [47]. When the RMB is more widely used and held, and its international recognition increases, the international attributes of its financial assets can be further enhanced, which is an important way to enhance financial stability and RMB internationalization under open market conditions. Therefore, this study selects "RMB financial assets held by foreign institutions and individuals" as another quantitative indicator of the degree of RMB internationalization at the micro level, aiming to measure RMB internationalization from the perspective of residents' daily financial business. Fig 1 illustrates this.

Loans, deposits, bonds, and stocks are the four major categories of domestic RMB financial assets held by overseas institutions and individuals. As Fig 1 shows, driven by the opening up of the financial market in recent years, the scale of foreign investors' RMB financial assets has steadily increased. By the end of March 2022, foreign institutions and individuals held 9.8 trillion yuan of domestic RMB financial assets, of which stocks and bonds accounted for 32.5% and 40.4%, respectively. Compared to ten years ago, the structure is more optimized, which is reflected in the gradual shift from deposits and loans to securities. From the perspective of the individual businesses in international financial services trade in developed countries, securities business such as enterprise listings and bond issuances are the main business growth drivers, and have become key channels to improve the competitiveness of their financial services trade. Kawai [48] pointed out that the unshakable dominance of the US dollar and the rapid growth of the US economy have created a highly stable "dollar anchor," while the US dollar has created a relatively stable international exchange rate system. Under the current international monetary system, changing the anchoring effect of the RMB on the US dollar may be difficult in the short term. Therefore, an improvement of RMB internationalization will also indirectly

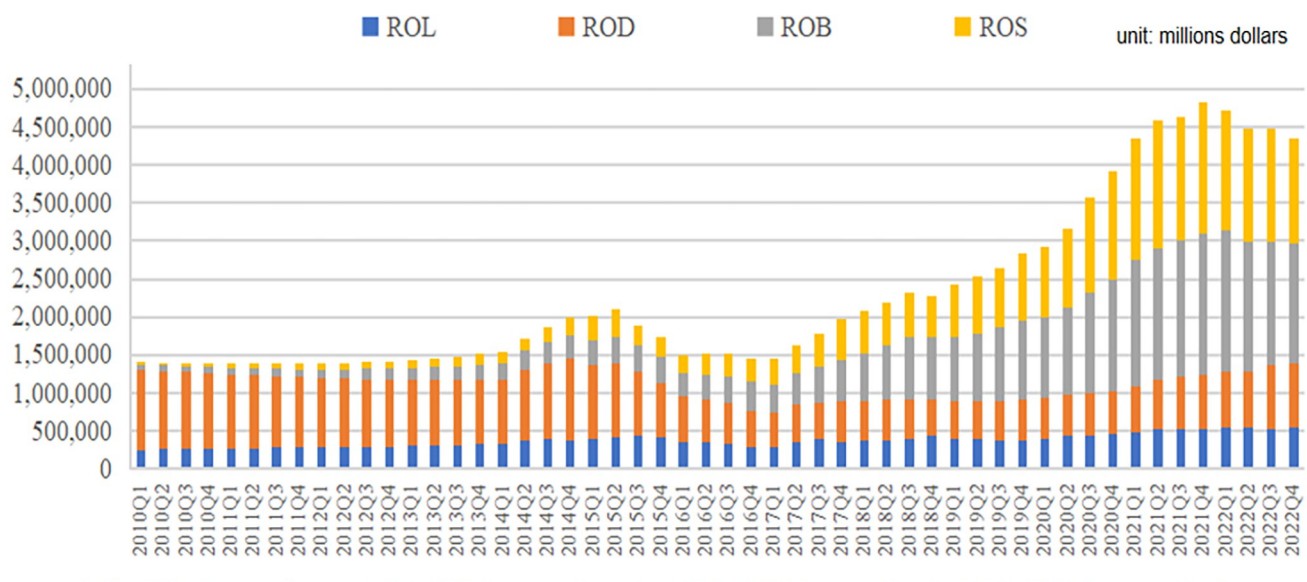

**Fig 1. RMB financial assets held by overseas institutions and individuals.**

increase the internationalization of the US dollar in the short term, thereby amplifying the relative advantage of the US in securities financial business. Conversely, for China, the expansion of securities financial businesses such as bonds and stocks, will not be conducive to narrowing the gap in financial services trade between China and the US compared with traditional deposit and loan businesses. Based on the above analysis, this study proposes hypothesis 2 as follows:

**Hypothesis 2:** *Different types of foreign institutions and individuals holding domestic RMB financial assets have heterogeneous promotion effects on China's narrowing of the gap between China and the US in financial services trade. Specifically, the promotion effect of traditional business, such as RMB overseas deposit and loan, may be stronger than that of holding RMB securities assets abroad.*

## 3.2 Model construction and variable definitions

**3.2.1 Model design.** First, to reflect the various influencing factors that affect the internationalization of the RMB, this study expresses this internationalization of the RMB as $A_i = f\ (ITSCR, FDICR, ROL, ROD, ROB, ROS, GDPCG, FGDPC, GTR)$. Second, the impact of RMB internationalization on China-US financial services trade is modeled as follows.

$$\ln Y = c + \sum p \ln A_i + \varepsilon_t \qquad (1)$$

To stabilize the data, and eliminate the influence of heteroscedasticity and dimensional differences, this study takes the logarithm of the time series data of all variables to reduce the variable scale (Except growth rate of trade in goods). In Eq (1), Y is the variable to be explained, c is a constant term, $A_i$ is the explanatory variable reflecting RMB internationalization (i = l, 2,. . .,9), and p is the estimated coefficients. Table 1 provides details on each variable.

In addition, considering potential mutual influence between indicators, this study chooses a nonstructural method, a VAR model, to fit the model of the relationship between variables. According to the theory of Sims and Toda [49], VAR models are highly practical and flexible

**Table 1. Variable definitions.**

| Variable name | Variable code |
|---|---|
| China-US Financial Services Trade Balance (CAFSTB) | Y |
| International Transaction Chinese-yuan Ratio (ITSCR) | A1 |
| Foreign Direct Investment Chinese-yuan Ratio (FDICR) | A2 |
| RMB Overseas Loan (ROL) | A3 |
| RMB Overseas Deposit (ROD) | A4 |
| RMB Offshore Bond Issuance (ROB) | A5 |
| RMB Overseas Stock Offering (ROS) | A6 |
| The ratio of China's GDP to Global GDP (GDPCG) | A7 |
| The Ratio of Added Value of Financial Industry to China's GDP (FGDPC) | A8 |
| Growth Rate of Trade in Goods (GTR) | A9 |

for multivariate time-series analyses. Particularly, due to the intricate relationship between macroeconomic variables with real-time linkage and hysteresis, the VAR model has become a basic paradigm in macroeconomic analyses. This model can effectively reveal the dynamic relationships between different variables, providing an important analytical tool for research in the fields of economics and finance.

Assuming that there is a relationship between $y_{1t}$ and $y_{2t}$, we can establish the following AR model:

$$y_{1,t} = f(y_{1,t-1}, y_{1,t-2}, \ldots) \tag{2}$$

$$y_{2,t} = f(y_{2,t-1}, y_{2,t-2}, \ldots) \tag{3}$$

We can use simultaneous form to solve the problem that the above equation cannot capture the dynamic relationship between the two variables. The VAR model is a system of simultaneous equations. Its structure has two parameters, the number of variables N and the maximum lag k. Assuming N = 2, k = 1, the VAR model is as follows:

$$\begin{aligned} y_{1,t} &= \mu_1 + \pi_{11.1} y_{1,t-1} + \pi_{12.1} y_{2,t-1} + u_{1t} \\ y_{2,t} &= \mu_2 + \pi_{21.1} y_{1,t-1} + \pi_{22.1} y_{2,t-1} + u_{2t} \end{aligned} \tag{4}$$

In the above, $\mu_{1t}, \mu_{2t} \sim IID(0, \sigma^2)$, $\mathrm{Cov}(\mu_{1t}, \mu_{2t}) = 0$. Let us transform it into the matrix form:

$$\begin{bmatrix} y_{1t} \\ y_{2t} \end{bmatrix} = \begin{bmatrix} \mu_1 \\ \mu_2 \end{bmatrix} + \begin{bmatrix} \pi_{11.1} & \pi_{12.1} \\ \pi_{21.1} & \pi_{22.1} \end{bmatrix} \begin{bmatrix} y_{1,t-1} \\ y_{2,t-1} \end{bmatrix} + \begin{bmatrix} u_{1t} \\ u_{2t} \end{bmatrix}$$

Suppose that, then, $Y_t = \begin{bmatrix} y_{1t} \\ y_{2t} \end{bmatrix}$, $\mu = \begin{bmatrix} \mu_1 \\ \mu_2 \end{bmatrix}$, $\Pi_1 = \begin{bmatrix} \pi_{11.1} & \pi_{12.1} \\ \pi_{21.1} & \pi_{22.1} \end{bmatrix}$, $ut = \begin{bmatrix} u_{1t} \\ u_{2t} \end{bmatrix}$, we can come to an equation:

$$Y_t = \mu + \Pi_1 Y_{t-1} + u_t$$

Therefore, the general VAR model containing N variables as well as k lags is:

$$Y_t = \mu + \Pi_1 Y_{t-1} + \Pi_2 Y_{t-2} + \Pi_3 Y_{t-3} + \ldots\ldots + \Pi_p Y_{t-k} + u_t, \ \mu_t \sim IID(0, \Omega) \tag{5}$$

In Eq (4), *Yt* is a column vecor of *N×1* ranks of time-series; that is $Y_t =$

$$\begin{bmatrix} LNCAFSTB \\ LNITSCR \\ LNFDICR \\ LNROL \\ LNROD \\ LNROB \\ LNROS \\ LNGDPCG \\ LNFGDPC \\ GTR \end{bmatrix},$$

$\Pi_1,...,\Pi_p$ represents the *N×N* corresponding coefficient matrix, *μ* is a coefficient column vector of *N×1* ranks, $\mu_t \sim IID(0,\Omega)$ is a column vector of *N×1* ranks of random errors, and k represents the order of the endogenous variable lag. On the right of each equation of VAR model, there are only lags of endogenous variables which are not correlated with $\mu_t$, Therefore, we can get consistent estimates for parameters with OLS method estimating each equation. Since the VAR model is essentially a model that describes the dynamic relationship between endogenous variables, we add both of the control variables and other explanatory variables to the model as endogenous variables.

**3.2.2 Variable definitions.** The explained variable is China-US Financial Services Trade Balance (CAFSTB). The sample data show China has been in a deficit position for a long time in China-US trade in financial services. Furthermore, the change of the import and export ratio of financial services trade between them can reflect the evolution of the country's position in the financial services trade. Therefore, this study uses the ratio of import and export of financial services trade between China and the US to measure the balance in financial services trade between the two countries: *CAFSTB = CAFST/ACFST*, where *CAFST* and *ACFST* are China's financial services exports and imports to the US, respectively.

Several explanatory variables are used. First, this study uses International Transaction Chinese-yuan Ratio (ITSCR) and FDI Chinese-yuan Ratio (FDICR). These two indicators are selected based on the calculation of the RII, as described in the "Report on RMB Internationalization" issued by Renmin University of China. The report provides a measure of the degree of RMB internationalization from the perspective of the three major functions of an international currency. The comprehensive RII is obtained by weighting and averaging these indicators through principal component analysis. The indicator system used in this study is listed in Table 2.

Table 3 presents the final weights of the secondary indicators to measure the degree of RMB internationalization. The quarterly RII can be obtained by weighting these secondary indicators and averaging them to obtain the annual RII from 2010–2021 (Fig 2).

**Table 2. Measuring indicators of RMB internationalization.**

| Level 1 indicators | Level 2 indicators | Variables | Data Sources |
|---|---|---|---|
| Medium of exchange | International Transaction Chinese-yuan Ratio | ITSCR | World Bank, IMF, PBOC |
| | Foreign Direct Investment Chinese-yuan Ratio | FDICR | IMF、IFS |
| Unit of account | International bonds and notes Chinese-yuan Ratio | IBNCR | BIS |
| Store of value | Reserves Chinese-yuan Ratio | RCR | IMF |
| | Global foreign credit Chinese-yuan Ratio | GFCCR | BIS |

**Table 3. The final weight of each sub-indicator of RMB internationalization.**

| Indicators | FETCR | ITSCR | FDICR | IBNCR | RCR | GFCCR |
|---|---|---|---|---|---|---|
| Wights | 0. 1956 | 0. 1575 | 0. 1849 | 0. 1960 | 0. 1784 | 0. 1742 |

Since 2010, RMB internationalization has gone through initial, rapid development, downturn stage, and fluctuation promotion stages. From 2010 to 2011, the scope of RMB settlement for cross-border trade gradually expanded to the whole country, the cross-border use of the RMB increased steadily, and the internationalization of the RMB gradually started. From 2012 to 2015, China continued to improve the RMB internationalization infrastructure (authorizing RMB overseas clearing banks and building RMB cross-border payment system), strengthen currency cooperation with other economies in the world (signing bilateral currency swap agreements), and achieve a higher level of opening up (proposed the "Belt and Road" initiative). These efforts promoted the rapid growth of RMB transactions, the degree of RMB internationalization rose rapidly, and peaked in the third quarter of 2015. Then, affected by the "8.11" exchange rate reform and the continuous depreciation pressure on the RMB, the RII began to decline and reached a trough at the end of 2016. Since the beginning of 2017, with the gradual stabilization of the RMB exchange rate, implementation of the "Belt and Road" construction, and the opening of China's capital account, the degree of RMB internationalization has gradually picked up. However, affected by external factors, such as the China-US trade war, RII' s rise has fluctuated.

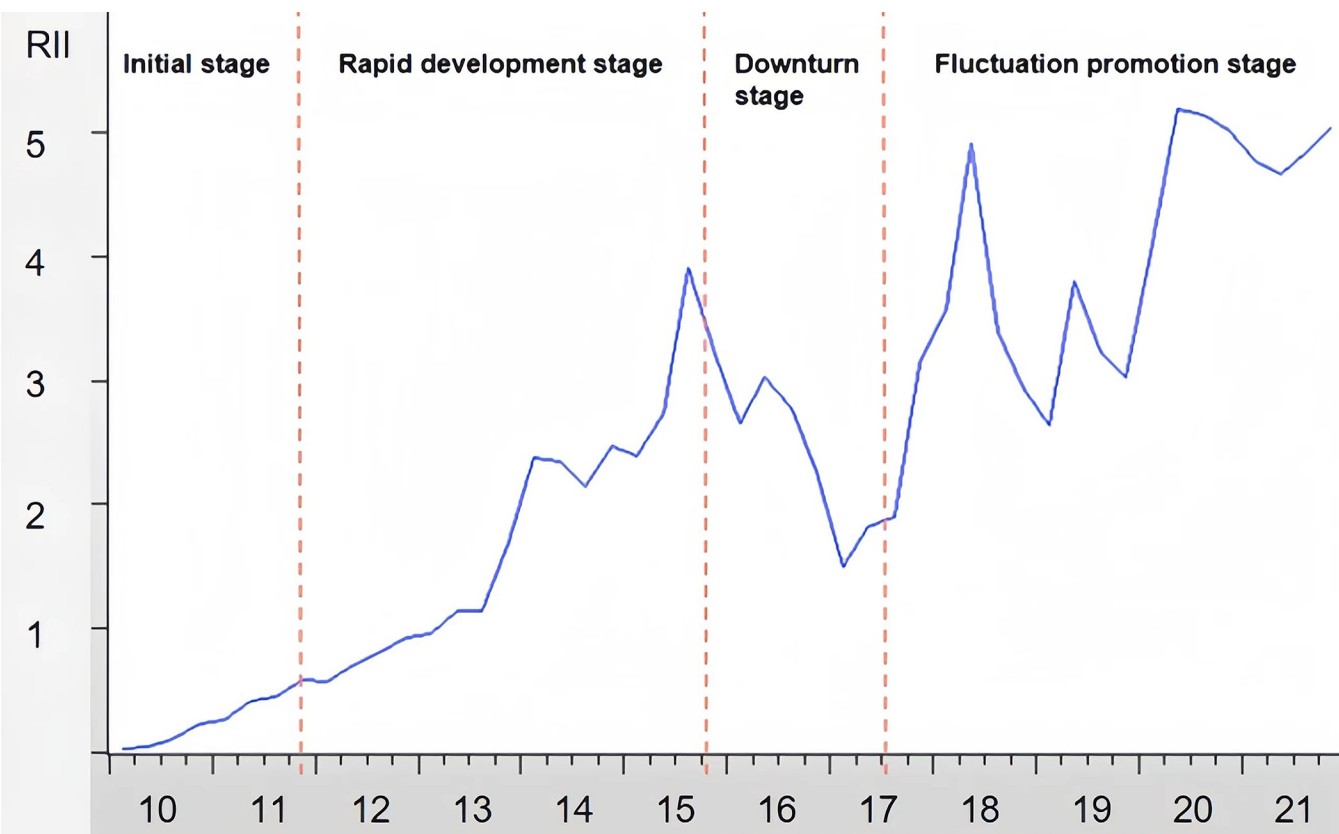

**Fig 2. RMB internationalization index.**

These observations in line with the basic economic situations in China, and it is quite reasonable to use the RII to reflect the degree of RMB internationalization. Based on the relevant literature and theoretical analysis, this study draws on the variable selection of the function of the international currency transaction medium. Specifically, the RMB share of international trade settlement and the RMB share of global direct investment are selected as the endogenous variables of this paper, which reflect the current level of RMB internationalization from the macro level.

The second of explanatory variables include those for overseas institutions and individuals held domestic RMB financial assets, mainly including loans, deposits, bonds, and stocks(ROL, ROD, ROB and ROS, respectively). As global investors continue to increase their interest in China's financial market, the RMB has gradually become a new safe haven investment and financing currency for various economic entities around the world. China began to provide RMB financial asset allocation solutions for global economic stability, financial market development, and cross-border capital flows in the "post-epidemic" era. Therefore, this study selects four types of foreign institutions and individuals holding domestic RMB financial assets as explanatory variables reflecting the degree of RMB internationalization at the micro-level, which is used to express the recognition and utilization rates of RMB financial assets in various economic entities around the world.

Finally, several control variables are used. First, to alleviate the possible endogeneity problem in the model caused by omitted variables, following Xiang [50], this study considers the availability of data and the loss of a certain degree of freedom due to the length of the lag period in the VAR system. Next, the following three control variables are added to the model. First, the level of economic development, measured by the proportion of China's GDP to the world's total GDP (GDPCG). A good economic foundation is a prerequisite for the development of financial services trade. The higher the level of economic development of a country, the stronger the competitiveness of its financial services trade. Second, the level of development of the financial industry, measured by the proportion of the added value of China's financial industry to its GDP (FGDPC). Financial services trade between countries often needs to be realized through financial institutions. The high-quality development of the financial industry can effectively improve the competitiveness of the financial services trade. Third, the scale of goods trade, measured by the growth rate of China's total imports and exports of goods trade (GTR). Financial services trade is inseparable from goods trade, which often drives the development of related financial services trade.

### 3.3 Data sources and descriptive statistics

The China-US Financial Services Trade Balance (CAFSTB) is measured from the ratio of China's exports to US financial services trade and its imports. Most of this indicator data are annual data. To ensure consistency of data frequency, the quadratic-match method is used to convert the annual data into quarterly data. The data are obtained from the UNCTAD.

Among the explanatory variables, the macro variables used to measure RMB internationalization include the RMB share of international trade settlements and the RMB share of global direct investment. The data comes from the 2010–2021 "RMB Internationalization Report" issued by the International Monetary Institute of Renmin University of China, the PBOC quarterly report on financial market operations, and the IMF database.

Four types of RMB assets held by foreign institutions and individuals are considered: overseas RMB deposits, loans, bonds, and stocks. The data are obtained from the 2010–2021 Statistical Yearbook of the PBOC. The standard deviation of securities financial assets (overseas RMB bonds and stocks) is significantly higher than that of traditional financial assets (deposits

**Table 4. Descriptive statistics results of each variables.**

| Variables | Obs. | Mean | Std. Dev. | Max. | Min. |
|---|---|---|---|---|---|
| China-US Financial Services Trade Balance (LNCAFSTB) | 48 | -1.7342 | 0.2931 | -1.0826 | -2.6751 |
| International Transaction Chinese-yuan Ratio (LNITSCR) | 48 | 13.9302 | 0.5446 | 14.6085 | 12.2699 |
| Foreign Direct Investment Chinese-yuan Ratio (LNFDICR) | 48 | 12.5507 | 1.2626 | 14.3040 | 8.9419 |
| RMB Overseas Loan (LNROL) | 48 | 12.7785 | 0.2011 | 13.1864 | 12.4461 |
| RMB Overseas Deposit (LNROD) | 48 | 13.4686 | 0.2889 | 13.8926 | 13.0015 |
| RMB Offshore Bond Issuance (LNROB) | 48 | 12.7554 | 0.9992 | 14.4284 | 11.0783 |
| RMB Overseas Stock Offering (LNROS) | 48 | 12.4963 | 1.1392 | 14.3715 | 10.5255 |
| The ratio of China's GDP to Global GDP (GDPCG) | 48 | -1.9885 | 0.2385 | -1.5738 | -2.4589 |
| The Ratio of Added Value of Financial Industry to China's GDP (FGDPC) | 48 | -2.6016 | 0.1416 | -2.2806 | -2.8747 |
| Growth Rate of Trade in Goods (GTR) | 48 | 2.2625 | 5.7705 | 20.3000 | -16.3000 |

and loans). This shows that securities financial assets are highly liquid and are more affected by fluctuations in the global financial market, which is reasonable.

Each control variable are calculated based on data from the statistical yearbook published by the National Bureau of Statistics of China and the China Stock Market & Accounting Research database.

The specific descriptive statistics results of each variables are shown in Table 4.

## 4. Empirical analysis

### 4.1 Stationary test of time series variables

The premise of establishing a credible time-series VAR model is to ensure the stability of each variable, thus preventing the problem of pseudo-regression in the model. Therefore, before modeling, the time-series data of the the aforementioned variables are first tested for stationarity. The Augmented Dickey-Fuller (ADF) method is used to test the unit root of each variable. The results are shown in Table 5.

After the first-order difference in the data series, the ADF test values of each variable are less than the critical value of 5% and the corresponding P values are all less than 0.05. Therefore, all variables pass the ADF stationarity test after the first-order difference. The integration

**Table 5. ADF stationarity test results for each variable.**

| Variables | difference order | Inspection type | ADF value | significance level | | | Prob.* | Stationarity |
|---|---|---|---|---|---|---|---|---|
| | | | | 1% | 5% | 10% | | |
| LNCAFSTB | 1 | (C,T,1) | -4.1709 | -4.1923 | -3.5208 | -3.1913 | 0.0106 | Yes |
| LNITSCR | 1 | (C,T,1) | -7.5124 | -4.1706 | -3.5107 | -3.1855 | 0.0000 | Yes |
| LNFDICR | 1 | (C,T,1) | -9.6539 | -4.1706 | -3.5107 | -3.1855 | 0.0000 | Yes |
| LNROL | 1 | (C,T,1) | -5.7013 | -4.1706 | -3.5107 | -3.1855 | 0.0001 | Yes |
| LNROD | 1 | (C,T,1) | -4.2307 | -4.1706 | -3.5107 | -3.1855 | 0.0085 | Yes |
| LNROB | 1 | (C,T,1) | -4.3218 | -4.1706 | -3.5107 | -3.1855 | 0.0067 | Yes |
| LNROS | 1 | (C,T,1) | -6.0069 | -4.1706 | -3.5107 | -3.1855 | 0.0000 | Yes |
| LNGDPCG | 1 | (C,T,1) | -3.3210 | -4.1865 | -3.5181 | -3.1897 | 0.0765 | Yes |
| LNFGDPC | 1 | (C,T,1) | -3.7012 | -4.1985 | -3.5236 | -3.1929 | 0.0335 | Yes |
| GTR | 1 | (C,T,1) | -9.4926 | -4.1756 | -3.5131 | -3.1869 | 0.0000 | Yes |

Note: The ADF test type is (c, t, k), where c represents the intercept item, t represents the trend item, and k represents lag order.

order of each variable is the same, that is, the first-order integrated sequence, which can be further tested for cointegration.

## 4.2 Determination of optimal lag order for VAR model

The prerequisite for establishing a VAR model is not only to satisfy the stationarity condition but also to correctly determine the lag period k. If there are too few lag periods, the autocorrelation can be severe, leading to inconsistent parameter estimates. Appropriately increasing the k value (increasing the number of lagged variables) in the VAR model can eliminate the autocorrelation in the variables; however, the k value should not be excessively large. An excessive value of k causes the degrees of freedom to decrease, which directly affects the effectiveness of the model parameter estimator. This study uses the Likelihood Ratios Statistic(LR), Akaike Information Criterion (AIC), Schwartz Criterion (SC) criteria to select the lag order. Considering the number of sequences and sample size, the maximum lag order selected in this study is three.

(1) The k value is selected using the LR statistic, as follows:

$$LR = -2(\log L_{(k)}) - \log L_{(k+1)}) \sim X^2(N^2)$$

where logL (k) and logL (k+1) are the maximum likelihood estimates of the VAR (k) and VAR (k+1) models, respectively, and k represents the maximum lag period of the lag variable in the VAR model. When an increase in the lag period of the VAR model does not result in a significant increase in the value of the maximum likelihood function, or when the value of the LR statistic is less than the critical value, the newly added lag variable is meaningless to the VAR model.

(2) The k value is selected using the AIC

$$AIC = \log(\frac{\sum_{t=1}^{T} \hat{\mu}_t^2}{T}) + \frac{2k}{T}$$

Where is the residual, T is the sample size, and k is the maximum lag period. The principle behind selecting the k value is to minimize the AIC value while increasing the k value.

(3) The k value is selected using the SC

$$SC = \log(\frac{\sum_{t=1}^{T} \hat{\mu}_t^2}{T}) + \frac{k \log T}{T}$$

The results are summarized in Table 6. When the lag order is three, the values of the LR, AIC, and SC reach a minimum. Therefore, the third lag order is optimal and chosen. Then, a VAR (3) model is established for each variable. The structure is shown in (3), where represents

**Table 6. Test values for determining the lag order of the VAR equation.**

| Lag | LogL | LR | FPE | AIC | SC | HQ |
|---|---|---|---|---|---|---|
| 1 | 457.8906 | NA | 6.31e-20 | -15.9063 | -11.7914 | -14.4096 |
| 2 | 582.5278 | 138.4858 | 3.81e-20 | -17.0012 | -8.9716 | -14.0079 |
| 3 | 837.7622 | 170.1562* | 3.12e-22* | -23.9005* | -11.8561* | -19.4105* |

the corresponding coefficient matrix and represents the random disturbance terms:

$$
\begin{bmatrix}
LNCAFSTB_t \\
LNITSCR_t \\
LNFDICR_t \\
LNROL_t \\
LNROD_t \\
LNROB_t \\
LNROS_t \\
LNGDPCG_t \\
LNFGDPC_t \\
GTR_t
\end{bmatrix}
= \prod
\begin{bmatrix}
LNCAFSTB_{t-1} \\
LNITSCR_{t-1} \\
LNFDICR_{t-1} \\
LNROL_{t-1} \\
LNROD_{t-1} \\
LNROB_{t-1} \\
LNROS_{t-1} \\
LNGDPCG_{t-1} \\
LNFGDPC_{t-1} \\
GTR_{t-1}
\end{bmatrix}
+ \prod
\begin{bmatrix}
LNCAFSTB_{t-2} \\
LNITSCR_{t-2} \\
LNFDICR_{t-2} \\
LNROL_{t-2} \\
LNROD_{t-2} \\
LNROB_{t-2} \\
LNROS_{t-2} \\
LNGDPCG_{t-2} \\
LNFGDPC_{t-2} \\
GTR_{t-2}
\end{bmatrix}
+ \prod
\begin{bmatrix}
LNCAFSTB_{t-3} \\
LNITSCR_{t-3} \\
LNFDICR_{t-3} \\
LNROL_{t-3} \\
LNROD_{t-3} \\
LNROB_{t-3} \\
LNROS_{t-3} \\
LNGDPCG_{t-3} \\
LNFGDPC_{t-3} \\
GTR_{t-3}
\end{bmatrix}
+
\begin{bmatrix}
\mu_{1t} \\
\mu_{2t} \\
\mu_{3t} \\
\mu_{4t} \\
\mu_{5t} \\
\mu_{6t} \\
\mu_{7t} \\
\mu_{8t} \\
\mu_{9t} \\
\mu_{10t}
\end{bmatrix}
\tag{6}
$$

### 4.3 Granger causality test

Table 7 shows the Granger causality test between China-US financial services trade and the nine other variables. At the 10% significance level, the hypothesis "the level of RMB internationalization is not the cause of China-US financial services trade" is rejected. Therefore, RMB internationalization is the Granger cause of China-US financial services trade. Changes in the level of economic development, the development level of the financial industry, and the scale of goods trade impact changes in the degree of RMB internationalization. These test results are consistent with economic theory and real experience.

### 4.4 Cointegration test

To verify whether there is a long-term equilibrium relationship between the variables, this study uses the Johansen cointegration method. The test results are listed in Table 8.

The trace statistics are all greater than the critical value of 0.05, and the P-value is less than 0.05; thus, there is a cointegration relationship between the variables at the 5% significance level. There are five cointegration relationships, indicating a long-term stable equilibrium relationship between the variables.

### 4.5 Stability test of VAR model

The judgment of model stability is key link to determining whether the VAR model is used reasonably. Therefore, before establishing a VAR model, its stability must be tested. In this study, an AR test is used to determine the stability of the model, and the test results are shown

**Table 7. Granger causality test results.**

| Excluded | Chi-sq | df | Prob.* |
|---|---|---|---|
| LNFDICR | 7.424964 | 3 | 0.0595 |
| LNITSCR | 7.002995 | 3 | 0.0718 |
| LNROB | 13.32212 | 3 | 0.0040 |
| LNROD | 6.08475 | 3 | 0.0076 |
| LNROL | 5.78477 | 3 | 0.0926 |
| LNROS | 12.36427 | 3 | 0.0062 |
| LNGDPCG | 2.85415 | 3 | 0.0911 |

**Table 8. Cointegration test results.**

| Hypothesized No. of CE(s) | Eigenvalue | Trace statistic | 0.05 Critical Value | Prob.* |
|---|---|---|---|---|
| None * | 0.797025 | 362.1290 | 239.2354 | 0.0000 |
| At most 1 * | 0.733436 | 288.7741 | 197.3709 | 0.0000 |
| At most 2 * | 0.717132 | 227.9557 | 159.5297 | 0.0000 |
| At most 3 * | 0.702403 | 169.8680 | 125.6154 | 0.0000 |
| At most 4 * | 0.567652 | 114.1153 | 95.75366 | 0.0015 |
| At most 5 * | 0.505866 | 75.54314 | 69.81889 | 0.0162 |

Note: * Rejection of the null hypothesis at the 5% significance level.

in Fig 3. The stability judgment of the VAR model is primarily based on whether the characteristic roots of the matrix are in a unit circle. If the characteristic root of the matrix falls within a unit circle, the VAR model is stable. Indeed, each characteristic root falls within the unit circle; that is, the module corresponding to each unit root is less than or equal to one. This result shows that the VAR model is stable, which also lays the foundation for the subsequent impulse response and variance decomposition.

## 4.6 Impulse response function analysis

To better analyze the actual economic significance of the VAR model, it is necessary to analyze its impulse response function. The impulse response function analyses the response of the endogenous variable to the changes in a disturbance item. When the model is impacted, other variables are affected through system conduction. This study analyzes the impact of international settlement Chinese-yuan ratio, FDI Chinese-yuan ratio, and four types of RMB assets on China-US financial services trade. Fig 4 shows their impact.

The impact of the six variables on LNCAFSTB has the following two common characteristics: (1) In the early and middle stages of the response, these variables have varying degrees of negative effects on CAFSTB. (2) In the long term, the impact turns into a relatively significant positive effect. This shows that the improvement in RMB internationalization reduces CAFSTB in the early and mid-term responses, implying that it will reduce China's competitiveness in China-US financial services trade in the short term. However, a significant improvement in the ratio is observed thereafter. This is because under the current international monetary system dominated by the US dollar, there will be an anchoring effect between the RMB and the US dollar. Thus, the increase in RMB international settlements and RMB asset holdings indirectly increases the currency strength of the US dollar for a certain period. In addition, the financial industry infrastructure of the US has a natural comparative advantage position. Thus, the impact of various RMB internationalization indicators will further amplifies the advantage effect of the US in the initial stage. Consequently, the financial industry of the US will increase its RMB-related business volume in the short term by virtue of its advantages, and thus, the export volume of financial services to China will increase accordingly. However, in the long run, an improvement in the level of RMB internationalization will gradually show a positive effect on China's financial services trade competitiveness. This shows that an improvement in the level of RMB internationalization can promote the competitiveness of China's financial services trade, but it will take a long time to achieve this. Based on the real environment and economic principles, the main reasons are as follows:

① Facilitate cross-border transactions: As the level of RMB internationalization increases, the RMB is used more widely in international trade and investment, especially in trade and

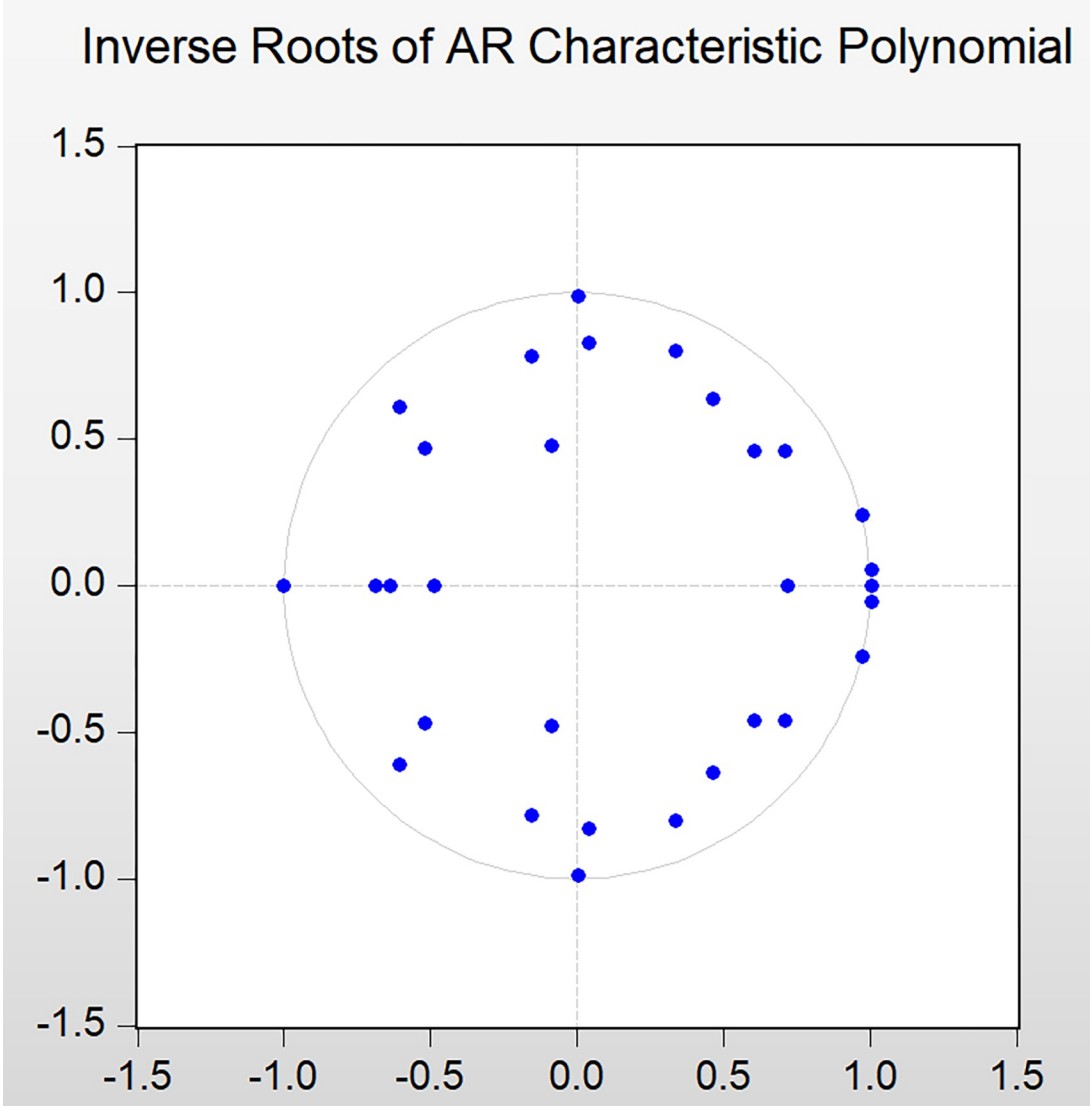

**Fig 3. Distribution of characteristic roots of VAR model.**

investment activities related to China. This will make it easier for Chinese companies and individuals to use the RMB for cross-border transactions, reduce dependence on the US dollar, and thus, promote financial services trade between China and the US.

② Reduce transaction costs: RMB internationalization can reduce the costs of cross-border transactions, including foreign exchange conversion fees and exchange risks. This makes Chinese companies more competitive and attracts more international customers to choose China's financial services, thereby narrowing the financial services trade gap between China and the US.

③ Increase the competitiveness of financial institutions: RMB internationalization will also enhance the status and competitiveness of Chinese financial institutions at the international stage, attracting more international customers to choose services provided by Chinese financial institutions. This will help Chinese financial institutions to expand their overseas

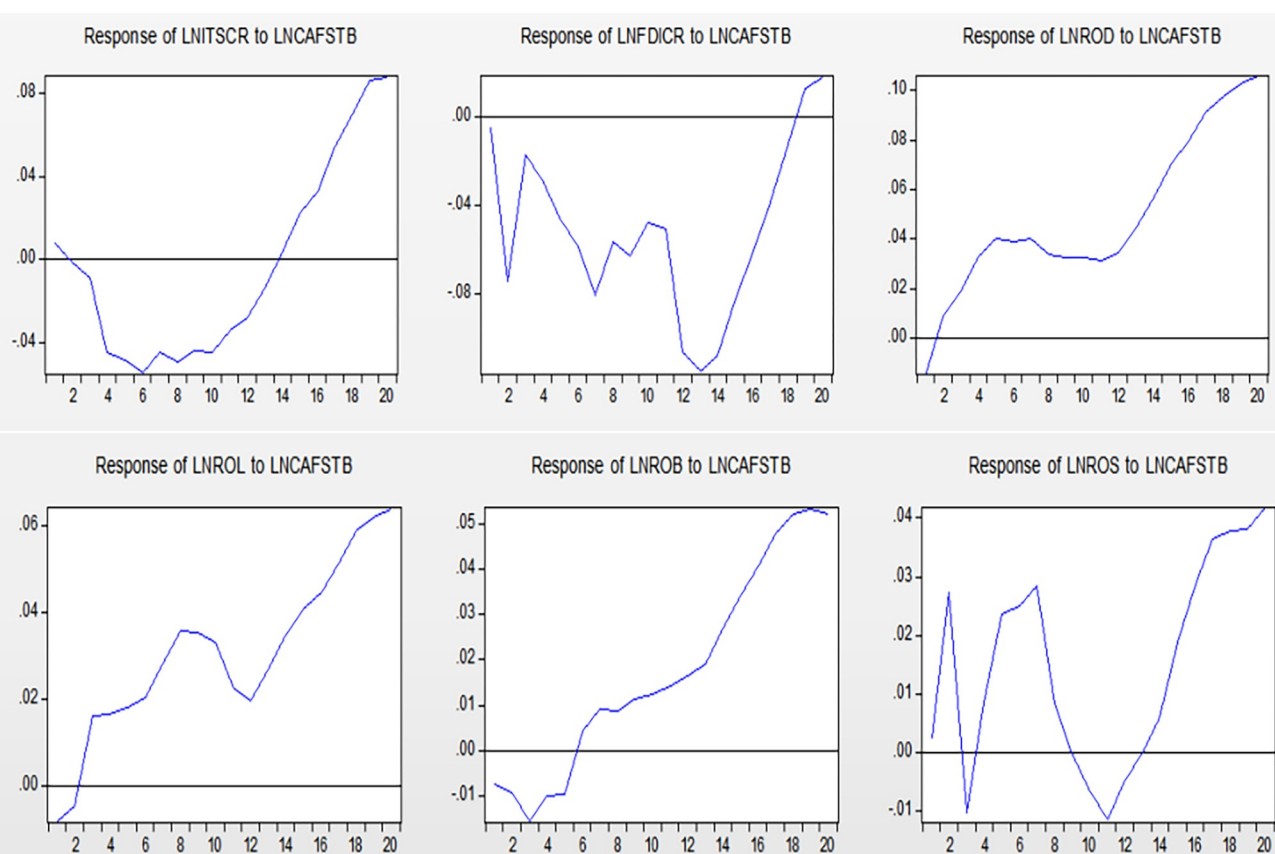

**Fig 4. Impulse response analysis diagram.**

business scale, thereby narrowing the gap in financial services trade between China and the US.

④ Promote financial market reforms: The advancement of RMB internationalization requires the reform and opening up of China's financial market, which will make it more open and transparent, improve the quality and efficiency of financial services, and attract more international customers and funds to the Chinese market.

From the time dimension of the impact and the specific impact degree, we find: (1) In the short term, the positive impact of ITSCR and FDICR on the CAFSTB is much smaller than the negative impact. FDICR has a negative effect in the first period of the impact and falls to a maximum of more than 0.1 after a short rebound. This shows that due to the anchoring effect of the RMB and US dollar, the increase in the RMB shares of international settlements and FDI will expand the advantageous effect of the US dollar under the current international monetary system. This means that it will have a significant reduction effect on the CAFSTB in the short term.

(2) From the perspective of medium-term impact, the impact of RMB assets held by the four types of overseas institutions or individuals on China-US financial services trade first exhibits a short-term negative impact, but the degree of this impact does not exceed 0.02. It then quickly becomes a positive shock, and the degree of the shock is significantly greater than that of the negative shock.

(3) In the long term, the six categories of indicators reflecting the degree of RMB internationalization have significant positive effects on the China-US financial services trade ratio. Thus, an improvement in the level of RMB internationalization enhances China's financial services trade competitiveness in the long run. At the macro level, the positive impact of ITSCR on CAFSTB is significantly greater than that of FDICR, exceeding 0.08. At the micro level, the positive impact of traditional RMB financial assets (ROD and ROL) on CAFSTB reaches 0.06–0.1. Compared with securities RMB financial assets (ROB and ROS), the positive impact is more significant. This also illustrates that the natural advantages of the US in securities financial services are more significant. Therefore, the increase in the internationalization of the RMB at this stage will enhance the US' leading position in China's financial services trade in securities financial services. Based on the relevant literature and observing the current international economic situation, we can draw the following main reasons for why the US' securities financial business is stronger than China's today:

① Mature markets and systems: The US securities market and financial system are very mature and complete, with highly developed stock exchanges, regulatory agencies, and legal systems, providing a good environment for securities financial businesses. In contrast, China's securities market and financial systems still have gaps in development and can be further improved.

② Rich financial products and tools: The US securities market provides a variety of financial products and tools, including stocks, bonds, futures, and options, to meet the needs of different investors. The richness of these products and instruments makes securities financial businesses in the US more diverse and flexible.

③ High degree of internationalization: The US securities market is one of the largest securities markets in the world, attracting a large number of international investors and funds. US financial institutions are highly competitive in the international market, and can provide a wide range of securities and financial services to global customers.

④ Strong innovation capabilities: The US financial market has always been famous for innovation, constantly launching new financial products and services to meet market demand and drive industrial development. This innovation capability provides a continuous impetus for the development of US securities financial businesses.

⑤ The capital market is large: The US capital market is large, with the largest stock exchange and the most listed companies in the world. This provides a wealth of investment opportunities and financing channels, and promoting the prosperity and development of securities financial businesses.

Notably, with the reform, opening up, and development of China's financial market, China's securities financial business is also constantly strengthening. In the future, this can help it narrow its gap with the US and gradually improve its international competitiveness.

Finally, comparing the impact of RMB offshore securities assets, ROS is highly volatile. This phenomenon is also consistent with the characteristics of stock assets, which are more susceptible to factors such as market sentiment, liquidity, and global economic conditions. Overall, the results are in line with basic economic principles and have strong practical significance.

## 4.7 Variance decomposition analysis

Another way to study the dynamics that exist between VAR variables is through forecast error variance decomposition. Like impulse response analysis, variance decomposition allows

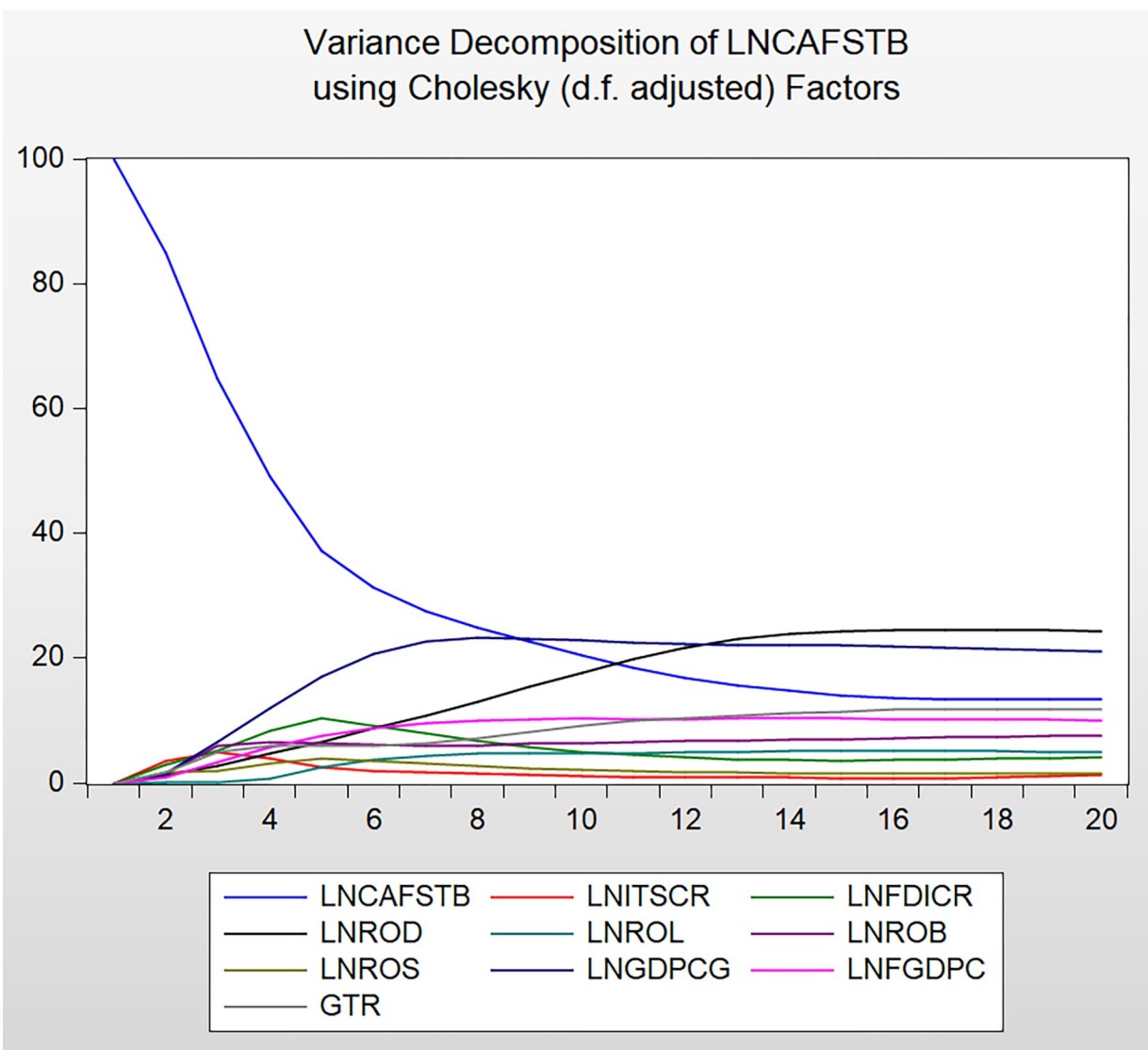

**Fig 5. Variance decomposition analysis chart.**

analyzing the behavior of endogenous variables following structural shocks and measuring the relative involvement of each shock on the variable to be measured. This method can reflect the relatively important information of each random disturbance that affects the endogenous variables in the system. The variance decomposition results can be used to analyze the contribution of ITSCR, FDICR, ROL, ROD, ROB, and ROS to the changes in the China-US financial services trade balance, as shown in Fig 5 and Table 9.

As the number of periods increases, the variance of the difference in CAFSTB explained by its own changes gradually decreases to approximately 13% after 15 periods. The part explained by the variance changes in the six indicators reflecting the level of RMB internationalization gradually increases, peaking after the 15th period, and remains at a level of approximately 87%. This shows that approximately 87% of the variance in CAFSTB is explained by the variance in the RMB internationalization level. This demonstrates that the variables selected in the model provide a strong explanation for the changes in the financial services trade between China and the US, and have good economic reference value.

**Table 9. Variance decomposition results.**

| Period | S.E. | LNCAFSTB | LNITSCR | LNFDICR | LNROD | LNROL | LNROB | LNROS | LNGDPCG | LNFGDPC | GTR |
|---|---|---|---|---|---|---|---|---|---|---|---|
| 1 | 0.034328 | 100 | 0 | 0 | 0 | 0 | 0 | 0 | 0 | 0 | 0 |
| 2 | 0.062134 | 85.00003 | 3.470651 | 2.926584 | 1.385514 | 0.046101 | 1.514792 | 1.712601 | 1.452373 | 0.849532 | 1.641823 |
| 3 | 0.087007 | 64.79521 | 4.839693 | 5.061595 | 2.646408 | 0.123147 | 5.847484 | 1.862923 | 6.604716 | 3.337182 | 4.881643 |
| 4 | 0.117993 | 49.10106 | 3.845576 | 8.34735 | 4.677527 | 0.725349 | 6.504262 | 3.16835 | 11.95848 | 5.690458 | 5.981595 |
| 5 | 0.148338 | 37.2742 | 2.475474 | 10.29213 | 6.468243 | 2.597402 | 6.384933 | 3.875503 | 16.9454 | 7.645233 | 6.041477 |
| 6 | 0.17516 | 31.25881 | 1.842418 | 9.25927 | 8.865147 | 3.77445 | 6.154348 | 3.570986 | 20.58525 | 8.772474 | 5.916852 |
| 7 | 0.200672 | 27.59941 | 1.6353 | 8.008261 | 10.71688 | 4.394144 | 5.963396 | 3.187264 | 22.61281 | 9.595184 | 6.287355 |
| 8 | 0.220729 | 24.86331 | 1.445059 | 6.660933 | 12.96811 | 4.755747 | 6.012627 | 2.734481 | 23.34463 | 10.06721 | 7.147894 |
| 9 | 0.23849 | 22.64186 | 1.238914 | 5.708929 | 15.44431 | 4.688893 | 6.268501 | 2.374605 | 23.10524 | 10.26381 | 8.26493 |
| 10 | 0.254434 | 20.4391 | 1.088523 | 5.016074 | 17.7276 | 4.749591 | 6.446411 | 2.13346 | 22.88869 | 10.3054 | 9.205163 |
| 11 | 0.26875 | 18.51671 | 0.976436 | 4.50757 | 19.95445 | 4.799513 | 6.610347 | 1.93022 | 22.52579 | 10.2797 | 9.899277 |
| 12 | 0.281725 | 16.93277 | 0.894527 | 4.10234 | 21.76059 | 4.844587 | 6.726022 | 1.784642 | 22.27693 | 10.26556 | 10.41203 |
| 13 | 0.292813 | 15.6887 | 0.854364 | 3.810403 | 23.01753 | 4.998922 | 6.816892 | 1.665026 | 22.1173 | 10.28736 | 10.7435 |
| 14 | 0.302138 | 14.73657 | 0.820278 | 3.631791 | 23.80655 | 5.057319 | 6.91841 | 1.564002 | 22.05738 | 10.28473 | 11.12297 |
| 15 | 0.309742 | 14.02571 | 0.790882 | 3.547506 | 24.25772 | 5.10146 | 7.033337 | 1.490283 | 21.99502 | 10.30373 | 11.45435 |
| 16 | 0.315385 | 13.58592 | 0.765085 | 3.643895 | 24.47234 | 5.115593 | 7.145326 | 1.454435 | 21.83791 | 10.28077 | 11.69873 |
| 17 | 0.319268 | 13.37259 | 0.780528 | 3.794964 | 24.56339 | 5.089818 | 7.277516 | 1.438456 | 21.62614 | 10.23833 | 11.81827 |
| 18 | 0.321871 | 13.34809 | 0.866601 | 3.909147 | 24.53057 | 5.051852 | 7.378699 | 1.432026 | 21.44183 | 10.17066 | 11.87053 |
| 19 | 0.323838 | 13.40171 | 1.073841 | 4.019364 | 24.42428 | 4.997817 | 7.475376 | 1.433157 | 21.23909 | 10.09335 | 11.84202 |
| 20 | 0.325392 | 13.47213 | 1.393496 | 4.066032 | 24.27589 | 4.951147 | 7.563797 | 1.431342 | 21.05409 | 10.01681 | 11.77528 |

Cholesky Ordering: LNCAFSTB LNITSCR LNFDICR LNROD LNROL LNROB LNROS LNGDPCG LNFGDPC GTR

The specific variance decomposition values in Table 9 show that the contribution of indicators of the level of RMB internationalization (ROD, ROL, ROB, and ROS) measured at the micro level to the variance changes in CAFSTB reach approximately 40% after the 15th period. This is much greater than the contribution of the indicators (ITSCCR, FDICR) measuring the level of RMB internationalization at the macro level to the variance of CAFSTB. This shows that the current RMB internationalization is still in the initial stage of development, and its impact on improving the competitiveness of China's financial services trade is concentrated at the level of RMB internationalization stakeholders and individuals' daily financial operations. This is because although the internationalization of the RMB has made some progress in the past few years, the US dollar still dominates the international monetary system. As the world's most important reserve and international trade settlement currency, it plays an irreplaceable role in the global financial system. The stability, liquidity, and credibility of the US dollar make it difficult for other currencies to compete.

Further analysis of the variance decomposition results of RMB assets (ROD, ROL, ROB, and ROS) held by overseas institutions and individuals shows that the contribution of traditional RMB overseas assets (ROD and ROL) to the variance in CAFSTB is much greater than that of securities RMB overseas assets. This also shows that the depth, openness, and diversity of financial products in the US financial market far exceed those of China, making the US more attractive in a highly volatile product market. Among the four types of financial assets, ROD contributes the most to the variance change in CAFSTB. RMB's current role in improving China's competitiveness in financial services trade is mainly concentrated in traditional financial services sectors with lower risks and is more biased towards overseas RMB deposit businesses. This is because for a currency that is still in its early stages of development, investors are more likely to use funds for investment or storage rather than expanding assets or

business scales through loans. Compared with the deposit business, overseas RMB loans usually involve more complex cross-border risks, including exchange rate risk and political risk, which makes some investors prefer a simple, low-risk deposit business instead of a loan business.

### 4.8 Robustness test

To ensure the robustness of the conclusions, this study replaces the different measurement indicators of RMB internationalization and again performs the impulse response analysis. The International Monetary Research Institute of Renmin University of China posits that with the orderly opening of the RMB capital account, the international monetary function of the RMB should be better reflected in the real economy. Specifically, it emphasizes the functions of the RMB as a currency for trade pricing and settlement, direct investment, and international bond transactions, and uses this to select appropriate variables and indicators to compile a comprehensive multivariable composite index, the RII. This is used to measure and reflect the level of RMB internationalization. Meanwhile, the PBOC has compiled the Cross-border RMB Index (CRI) by tracking the active use of the RMB in three key links: cross-border outflows, overseas transfers, and cross-border returns. This index transmits information on cross-border and overseas use and circulation of the RMB to customers in a more intuitive way, and more comprehensively reflects the development of RMB internationalization. These two indicators are relatively authoritative and comprehensive indicators for measuring the degree of RMB internationalization; therefore, they are selected for robustness testing. The impulse response analysis in Fig 6 shows that the impact of RII and CRI on the China-US financial services trade ratio (CAFSTB) experienced a short-term negative impact and gradually became a long-term positive impact. This finding is consistent with previous results and demonstrates their robustness.

## 5. Research conclusions and policy suggestions

This study uses the VAR model to investigate the impact of six indicators of RMB internationalization on China-US financial services trade balance: international trade settlement RMB ratio, FDI RMB ratio, overseas RMB loans, overseas RMB deposits, overseas RMB bonds, and overseas RMB stocks. Quarterly data from the first quarter of 2010 to the fourth quarter of 2021 are used, and Unit Root, Granger Causality, Cointegration, and Characteristic Root Tests are conducted on the established VAR model. Finally, impulse response and variance

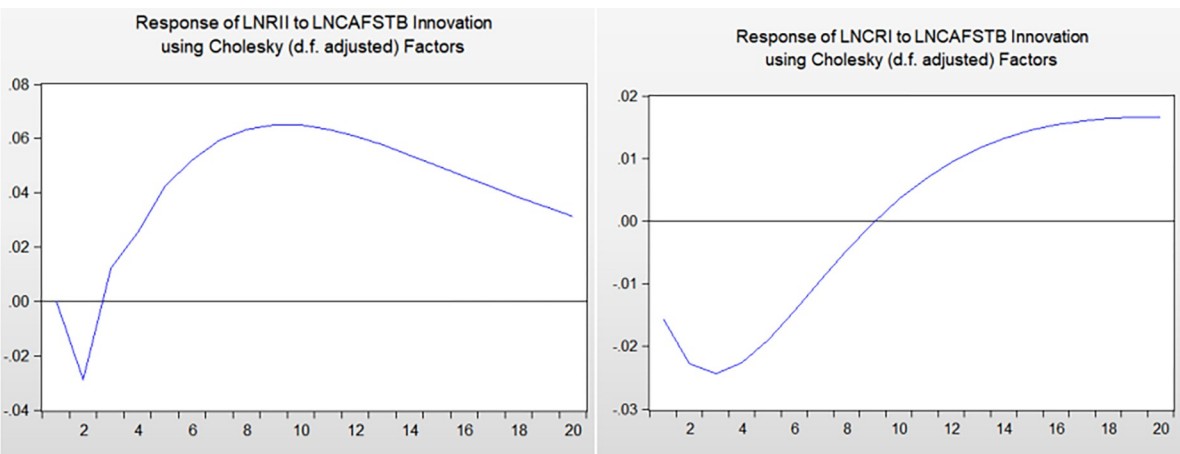

**Fig 6. Impulse response analysis chart after changing the RMB internationalization indicator.**

decomposition to are used study the impact and contribution of the six influencing factors on the ratio of China-US financial services trade. The conclusions are summarized below:

First, the impact of the six indicators on the China-US financial services trade varies in different periods. Specifically, since the existing monetary system is still based on the US dollar standard, the RMB should continue to have an anchoring effect with the US dollar for a long time, and the comparative advantage of the US in the financial services industry should remain stable for a long time. Therefore, an increase in the level of RMB internationalization can not only diminish China's financial service trade competitiveness in the short to medium term, but also amplify the advantage effect of the US. In other words, an increase in the internationalization of the RMB will enable the US to transform its natural advantages in the financial services industry into an increase in RMB business volume in the short term, thus continuing to widen the financial services trade balance between China and the US. However, in the long run, the results of the impulse response analysis show that an improvement in the level of RMB internationalization helps narrow the China-US financial services trade balance and the US. Thus, even in the context of the current US dollar currency system, improving the internationalization of the RMB can promote the competitiveness of China's financial services trade in the long-term. Hence, **hypothesis 1** is supported.

Second, among the four types of RMB assets held by overseas institutions and individuals, the impacts of overseas RMB loans and deposits, as traditional financial assets, and overseas RMB bonds and overseas RMB stocks, as securities financial assets, on China-US trade in financial services are diametrically opposed. During the sample statistical period, the negative impact of overseas RMB deposit and loan businesses is smaller than that of overseas RMB bonds and stocks businesses. Meanwhile, in the long term, the positive impact of the former is greater than the latter. Thus, the US has a more significant natural advantage in securities financial services than in traditional financial services. Therefore, at this stage, the development of overseas RMB deposits and loan services is more important for narrowing the gap in the financial services trade between China and the US. Therefore, **hypothesis 2** is supported.

Based on these conclusions, five policy implications are proposed for the development of China's RMB internationalization and financial services industry. First, although the level of RMB internationalization continues to improve, there is still much room for improvement. It is necessary to actively improve the role of the RMB in international economic exchanges, and to continuously expand the scope and areas of RMB use in international economic activities. To this end, the country should stabilize the inflation rate, build an RMB internationalization partner network, and enhance the international attractiveness of the RMB as a pricing unit and trading medium. Simultaneously, the RMB settlement process for cross-border trade should be further optimized. Enterprises should be encouraged to use the RMB for pricing and settlement, or provided more policy support when conducting cross-border trade.

Second, when conducting financial services trade with the US and other financial power-houses, Chinese financial services trade practitioners should mitigate their own mitigate their own weaknesses, and work on their strengths. China's financial market is still in the initial stages of development, and its financial infrastructure is imperfect. Thus, the country should first focus on developing the international competitiveness of the traditional RMB deposit and loan businesses, such as developing the offshore RMB market and adding more overseas branches of Chinese-funded banks to lay a solid foundation for China to enter more advanced international financial services in the future.

Third, China cannot ignore competition in the high value-added financial business market. This requires the Chinese government and relevant regulators to continue to improve the RMB internationalization infrastructure, promote the degree of opening up of China's

financial market, and further strengthen regulations and laws related to the financial market system and supervision, etc.

Fourth, the current international monetary system cannot drastically change in the short term, and the renminbi will be pegged to the US dollar for a long time. In this environment, China should realize the gap between itself and the traditional global financial powerhouses, such as the US, the United Kingdom, and Germany, and learn the strengths of these countries in the development of the financial industry for more than a hundred years. It should draw on and combine their advanced financial industry development concepts and methods according to its own conditions to open up a financial development path that suits China.

Fifth, China should achieve further improvements in the quality of economic development, level of RMB internationalization, and competitiveness of financial services trade. Specifically, China's economic development should follow a path of high-quality development that is innovative, coordinated, green, open, and shared. The RMB should follow a credible, stable, and universally recognized international path. Finally, China's financial services should follow a digital development path of quality improvement and enhanced competitiveness. In summary, the Chinese government should encourage the RMB to play a greater role in international trade settlements and international foreign exchange reserves, and promote China's high-quality financial services to the world.

## 6. Limitations and future research directions

First, owing to limitations in data acquisition and considering the problem of unifying the time scale of variables, the research sample period of this study only ranges from the first quarter of 2010 to the fourth quarter of 2021. Future researchers should select samples with a larger time frane to obtain more robust results. Second, the RMB measurement method is based on international monetary functions at the macro level, and the daily financial business of economic individuals at the micro level. Scholars should measure the RMB internationalization level from other perspectives as well. Third, the VAR model has problems such as difficulty in variable selection, overfitting, and lack of exogenous variables. Researchers could use other research methods and analytical logic to further explore the main mechanisms and paths through which RMB internationalization affects China's financial services trade competitiveness.

## Supporting information

**S1 Dataset.**
(XLSX)

## Acknowledgments

Firstly, I would like to thank my supervisor, Professor Tao Xionghua. His patient guidance and professional advice had a profound impact on me and enabled me to successfully complete this study. In addition, I would like to express my special thanks to my colleagues and friends for their selfless help and support during the research process. Their cooperation and encouragement helped me maintain confidence and courage in difficult times. Finally, I would like to thank all the experts, academics, and peers who reviewed this study and provided valuable comments. Their review made the content and structure of my study more rigorous and complete.

## Author Contributions

**Data curation:** Yufei Lei.

**Formal analysis:** Yufei Lei.

**Investigation:** Yufei Lei.

**Methodology:** Yufei Lei.

**Resources:** Yufei Lei.

**Software:** Yufei Lei.

**Writing – original draft:** Yufei Lei.

**Writing – review & editing:** Yufei Lei.

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
