## [Decision Letter · Decision Letter 0]

6 Feb 2024

PONE-D-23-42358Impact of RMB Internationalization on China's Competitiveness in Financial Services Trade based on VAR model: Evidence from China-USPLOS ONE

Dear Dr. LEI,

Thank you for submitting your manuscript to PLOS ONE. After careful consideration, we feel that it has merit but does not fully meet PLOS ONE’s publication criteria as it currently stands. 

In view of the referees’ feedback and my own reading of your paper, we do not consider your paper to meet the journal’s criteria for publication. We have particular concerns about the robustness of the methods and analysis and the contribution to the literature. We are confident that the issues identified could be resolved with a major revision, so we invite you to address the issues noted below and resubmit the manuscript for a new revision round.

We look forward to receiving your revised manuscript.

Kind regards,

Juan E. Trinidad Segovia, PhD

Section Editor

PLOS ONE

A clean copy of the edited manuscript (uploaded as the new *manuscript* file).

Reviewers' comments:

Reviewer's Responses to Questions

**Comments to the Author**

1. Is the manuscript technically sound, and do the data support the conclusions?

Reviewer #1: Yes

Reviewer #2: Partly

2. Has the statistical analysis been performed appropriately and rigorously? 

Reviewer #1: Yes

Reviewer #2: Yes

3. Have the authors made all data underlying the findings in their manuscript fully available?

Reviewer #1: Yes

Reviewer #2: Yes

4. Is the manuscript presented in an intelligible fashion and written in standard English?

Reviewer #1: No

Reviewer #2: Yes

5. Review Comments to the Author

Reviewer #1: 1. Author applies the VAR model to explore the potential impact of RMB internationalization on China’s competitiveness in China-US financial services trade. It is an interesting perspective to study RMB internationalization and its impacts.

2. Other determining factors of financial service trade other than RMB internationalization should be considered and included in the model as control variables.

3. Robustness test can be conducted by using other indicators of RMB internationalization measurement. It will improve reliability of empirical results.

4. Multicollinearity should be carefully considered in author’s VAR model since it is highly possible that there is multilinear among variables on level of RMB internationalization.

5. Minor errors: 4.1 and 4.2 titles are the same; English writing could be improved by academic editing.

Reviewer #2: This paper investigates the impact of RMB internationalization on China's competitiveness in financial services trade with the US. The authors build a VAR model based on time series data from 2010 to 2021 and analyze the effects of different indicators of RMB internationalization on financial services trade. The results show that RMB internationalization helps narrow the balance of financial services trade between China and the US, with a significant lag. The paper concludes with specific countermeasures for China to enhance its international competitiveness in the financial services sector. The paper is relatively clear and the structure is reasonable. Besides that, the conclusion has practical significance. However, the article needs further improvement in the following aspects.

1. The introduction need to be revised. The introduction gives too much background and requires only a concise paragraph. The author is too slow to get to the point and it detracts from the reading experience. The introduction of the article places too much emphasis the importance of the financial industry without discussing why the study was conducted. The introduction should explain the background meaning of this paper. Start with your research question identify your subject and discuss your motivation. Words like gap and first time need to be used with caution in the elaboration of marginal contribution.

2. The literature review need to be revised. You should discuss tell what is the new thing we learn from your paper that we previously didn’t know about. Furthermore, the literature review needs to be more adequate. It is suggested that the author refer to the following literatures and systematically reviewed them.

Li, Z., Mo, B., & Nie, H. (2023). Time and frequency dynamic connectedness between cryptocurrencies and financial assets in China. International Review of Economics & Finance, 86, 46-57.

Yang, C., Chen, L., & Mo, B. (2023). The spillover effect of international monetary policy on China’s financial market. Quantitative Finance and Economics, 7(4), 508-537.

Lin, Y. X., Chen, X., & Lan, H. Y. (2023). Analysis and prediction of American economy under different government policy based on stepwise regression and support vector machine modelling. Data Science in Finance and Economics, 3(1), 1-13.

3. The author has no relevant explanation for the reason for using the model. Furthermore, the paper lacks details on the implementation of the VAR model. It would be helpful to provide information on the lag length selection, model specification, and estimation techniques.

4. The paper does not discuss the limitations of the VAR model and potential sources of endogeneity or omitted variable bias.

5. This paper based on the time series data of China-US financial services trade from 2010 to 2021. However, the author said in the introduction that China has taken a series of measures to promote the internationalization of RMB since 2008. So why doesn’t the author use earlier data to compare the differences before and after the internationalization of RMB?

6. The economic explanation of the empirical results is insufficient. In this paper, each empirical result is only to explain whether the relevant test statistics are significant or not, but the explanation of economics is not enough. It is suggested that the graph and table involving the empirical results should be combined for economic explanation. In fact, statistical significance can only explain the "what", but does not explain the "why", that is, the paper needs to explain the "why" based on the econometric analysis.

6. PLOS authors have the option to publish the peer review history of their article (what does this mean?). If published, this will include your full peer review and any attached files.

Reviewer #1: No

Reviewer #2: No

---

## [Author Response · Author response to Decision Letter 0]

15 Mar 2024

Response letter

Title: Impact of RMB Internationalization on China's Competitiveness in Financial Services Trade based on VAR model: Evidence from China-US

Number.: PONE-D-23-42358

I express my profound gratitude to the esteemed editors and two diligent reviewers for their invaluable contributions in the form of constructive comments and suggestions. Their insightful feedback has played a pivotal role in improving the overall quality and scholarly merit of my manuscript. I have meticulously addressed each comment and suggestion in a detailed and methodical manner, offering comprehensive responses in the section below. Additionally, for your easy reference, all the main amendments in the revised paper are highlighted in yellow. Furthermore, the comments and recommendations provided by the editors and reviewers have been distinctly presented in italics, enabling effortless comprehension and reference.

Response to the Editor:

Editor point #1: Please ensure that your manuscript meets PLOS ONE's style requirements, including those for file naming. The PLOS ONE style templates can be found at：

Author response #1: I sincerely appreciate your valuable comments and suggestions provided for my manuscript. I have thoroughly reviewed my manuscript and meticulously updated the references in accordance with the prescribed Numbered reference style adopted by PLOS ONE. This rigorous adherence to the journal’s guidelines not only ensures compliance but also promotes consistency in the format and style across the entire manuscript. 

Editor point #2: We suggest you thoroughly copyedit your manuscript for language usage, spelling, and grammar. If you do not know anyone who can help you do this, you may wish to consider employing a professional scientific editing service.

Author response #2: I extend my sincere gratitude for your valuable feedback regarding my manuscript. In response to your insightful recommendations, I have taken the initiative to utilize the esteemed Editage(www.editage.com)’s Language Editing services, having placed an order with No.IJNRO_1. Details of this professional service and a copy of the manuscript with using track changes or highlighting have been uploaded as a zip file in Supporting Documents. Furthermore, I have diligently revised the manuscript, paying particular attention to rectifying any grammatical errors and enhancing the overall quality of the language used. Rest assured, the revised manuscript now aligns with the stringent standards set by PLOS ONE. I am confident that my efforts to improve the language quality will meet your expectations.

Editor point #3: Please include captions for your Supporting Information files at the end of your manuscript, and update any in-text citations to match accordingly. Please see our Supporting Information guidelines for more information: http://journals.plos.org/plosone/s/supporting-information.

Author response #3: Your suggestions have been immensely helpful in improving the quality and normativeness of my research. Taking into consideration your suggestion, I have incorporated a dedicated section titled “Support Information” (on page 35) right before the “References” section. The Supporting Information includes one dataset (hyperlink) containing all the data required for the models and tests of this article (click the hyperlink to open the Excel sheet). And, I have uploaded this data named "New Dataset" in the Supporting Information. The data availability statement is as follows: Data will be made accessible upon request. Besides, I have uploaded all the figures in the article to PACE.

In order to improve the readability of the article and ensure standardization, I carefully reviewed the citations in the article and made necessary adjustments and updates to the references based on your insightful suggestions.

Response to Reviewer 1:

Reviewer point #1: Author applies the VAR model to explore the potential impact of RMB internationalization on China’s competitiveness in China-US financial services trade. It is an interesting perspective to study RMB internationalization and its impacts.

Author response #1: I really appreciate your interest and comments on my research. This article is dedicated to finding ways for the Chinese government to improve the competitiveness of financial services trade in the context of RMB internationalization. I believe this research motivation is meaningful and necessary.

Reviewer point #2: Other determining factors of financial service trade other than RMB internationalization should be considered and included in the model as control variables.

Author response #2: I appreciate your insightful advice. Through further study of relevant literature[1][2] and in-depth analysis of the research content, I admit that this study should select other influencing factors of financial services trade as control variables and add them to the model. 

Following Xiang[3], this study considers the availability of data and the loss of a certain degree of freedom due to the length of the lag period in the VAR system. Thus, the following three control variables are added to the model. First, the level of economic development, measured by the proportion of China's GDP to the world's total GDP (GDPCG). A good economic foundation is a prerequisite for the development of financial services trade. The higher the level of economic development of a country, the stronger the competitiveness of its financial services trade. Second, the level of development of the financial industry, measured by the proportion of the added value of China's financial industry to its GDP (FGDPC). Financial services trade between countries often needs to be realized through financial institutions. The high-quality development of the financial industry can effectively improve the competitiveness of the financial services trade. Third, the scale of goods trade, measured by the growth rate of China's total imports and exports of goods trade (GTR). Financial services trade is inseparable from goods trade, which often drives the development of related financial services trade.

Through the ADF method, all the above control variables passed the stationarity test. The overall model remains stable after adding new control variables and the empirical results are consistent with the theoretical analysis. This approach can make the var model analyze the relationship between independent variables and dependent variables more comprehensively and accurately, and improve the reliability of research conclusions.

References

[1]Bernanke, B. S. Alternative explanations of the money-income correlation. In Carnegie-Rochester Conference Series on Public Policy. 1986;25:49-99 https://doi.org/10.3386/w1842

[2]Blanchard, O. J., & Quah, D. The dynamic effects of aggregate demand and supply disturbances.The American Economic Review. 1989;79:655-673. https://doi.org/10.3386/w2737

[3]Xiang YX, Wang XM. Research on the impact of currency internationalization on the competitiveness of financial services trade. Western Forum. 2022;04:41-54.

Reviewer point #3: Robustness test can be conducted by using other indicators of RMB internationalization measurement. It will improve reliability of empirical results.

Author response #3: I would like to express my gratitude for your insightful comments. To ensure the robustness of the conclusions, this study replaces the different measurement indicators of RMB internationalization and again performs the impulse response analysis (on page 32). The International Monetary Research Institute of Renmin University of China posits that with the orderly opening of the RMB capital account, the international monetary function of the RMB should be better reflected in the real economy. Specifically, it emphasizes the functions of the RMB as a currency for trade pricing and settlement, direct investment, and international bond transactions, and uses this to select appropriate variables and indicators to compile a comprehensive multivariable composite index, the RII. This is used to measure and reflect the level of RMB internationalization. Meanwhile, the PBOC has compiled the Cross-border RMB Index (CRI) by tracking the active use of the RMB in three key links: cross-border outflows, overseas transfers, and cross-border returns. This index transmits information on cross-border and overseas use and circulation of the RMB to customers in a more intuitive way, and more comprehensively reflects the development of RMB internationalization. 

These two indicators are relatively authoritative and comprehensive indicators for measuring the degree of RMB internationalization; therefore, they are selected for robustness testing. It can be seen from the impulse response analysis that the impact effects of these two indicators on CAFSTB are basically consistent with the previous empirical results. This finding further validates the robustness and explanatory power of the model.

Reviewer point #4: Multicollinearity should be carefully considered in author’s VAR model since it is highly possible that there is multilinear among variables on level of RMB internationalization.

Author response #4: I would like to express my sincere gratitude for your valuable feedback on my manuscript. Meanwhile, I fully understand the questions you raised and respect your rigorous academic vision. However, based on reviewing more relevant literature, I consider that the problem of multicollinearity may not need to be taken too seriously in VAR models, but more attention should be paid to the impulse response analysis and variance decomposition results.. 

According to Sims[1], Toda[2], and Lütkepohl’s[3] theory, in practical applications, the VAR (vector autoregressive) model usually does not need to worry too much about the problem of multicollinearity. This is mainly because the VAR model's own characteristics and modeling methods prevent multicollinearity from having a serious impact on it. Here are some explanations from these scholars on this issue:

(1) The variables in the VAR model are usually endogenous variables, and there is mutual influence between them, unlike the causal relationship in the traditional panel regression model. This endogeneity characteristic makes the VAR model less sensitive to multicollinearity.

(2) The VAR model assumes that variables are orthogonal, that is, there is no collinearity between them. This assumption enables the VAR model to better handle the relationship between variables. Even if there is a certain degree of correlation, it will not cause serious problems to the model.

(3) The VAR model is a dynamic model that takes into account the lag effect of time series data. Therefore, during the modeling process, lag terms are introduced to capture the dynamic relationship between variables, thereby reducing the impact of multicollinearity.

In addition, after reviewing other scholars’ research using VAR models, I found that there was no content or chapter that mentioned the multicollinearity. You may refer to the following literature. Although VAR models are less sensitive to multicollinearity than traditional regression models, some details still need to be paid attention to during the modeling process to ensure the robustness and accuracy of the model. For example, we should scientifically select the lag order, add appropriate control variables, improve the quality of time series data, etc. I would like to express my sincere thanks again for your suggestions. I will keep paying attention to VAR model methodology research, and carefully consider this problem in future study.

References

[1]Sims CA. Macroeconomics and reality. Econometrica: journal of the Econometric Society. 1980:1-48. https://doi.org/10.2307/1912017

[2]Toda HY, Yamamoto T. Statistical inference in vector autoregressions with possibly integrated processes. Journal of econometrics. 1995;66(1-2): 225-250. https://doi.org/10.1016/0304-4076(94)01616-8

[3]Lütkepohl H. New introduction to multiple time series analysis. Springer Science & Business Media. 2005. https://doi.org/10.1007/978-3-540-27752-1

[4]Ueda RM, Souza AM, Menezes RMCP. How macroeconomic variables affect admission and dismissal in the Brazilian electro-electronic sector: A VAR-based model and cluster analysis. Physica A: Statistical Mechanics and its Applications. 2020;557: 124872. https://doi.org/10.1016/j.physa.2020.124872

[5]Rodrigues SD, Ueda RM, Barreto AC, Zanini, RR, Souza AM. How atmospheric pollutants impact the development of chronic obstructive pulmonary disease and lung cancer: A var-based model. Environmental Pollution.2021;275: 116622. https://doi.org/10.1016/j.envpol.2021.116622

[6]Park C, Chung M, Lee S. The effects of oil price on regional economies with different production structures: A case study from Korea using a structural VAR model. Energy Policy.2011;39(12): 8185-8195. https://doi.org/10.1016/j.enpol.2011.10.018

[7]Brissimis SN, Magginas NS. Forward-looking information in VAR models and the price puzzle. Journal of Monetary Economics. 2006;53(6): 1225-1234. https://doi.org/10.1016/j.jmoneco.2005.05.014

[8]Liu S, Huang M, Yushui LI. Chinese agricultural insurance development in a VAR model. Procedia Computer Science. 2022;202: 399-407. https://doi.org/10.1016/j.procs.2022.04.056

Reviewer point #5: Minor errors: 4.1 and 4.2 titles are the same; English writing could be improved by academic editing.

Author response #5: I sincerely appreciate your advice on my manuscript. I have carefully checked the minor errors you mentioned and made corrections (on page 23). In response to editor’s insightful recommendations, I have taken the initiative to utilize the esteemed Editage(www.editage.com)’s Language Editing services. Furthermore, I have diligently revised the manuscript, paying particular attention to rectifying any grammatical errors and enhancing the overall quality of the language used.

I express my sincere gratitude for your valuable and perceptive comments. 

Response to Reviewer 2:

Reviewer point #1: The introduction need to be revised. The introduction gives too much background and requires only a concise paragraph. The author is too slow to get to the point and it detracts from the reading experience. The introduction of the article places too much emphasis the importance of the financial industry without discussing why the study was conducted. The introduction should explain the background meaning of this paper. Start with your research question identify your subject and discuss your motivation. Words like gap and first time need to be used with caution in the elaboration of marginal contribution.

Author response #1: I would like to express my gratitude for your invaluable feedback, which has significantly enhanced the quality of my research. In light of your suggestions, I acknowledge that the information could benefit from greater conciseness. Therefore, I deleted too much discussion of the importance of the financial industry and other redundant statements. Based on your suggestion, I introduce the research background of this article in the first paragraph. The second paragraph emphasizes the research theme of this article, RMB internationalization. In

---

## [Decision Letter · Decision Letter 1]

10 Apr 2024

Impact of RMB Internationalization on China's Competitiveness in Financial Services Trade based on the VAR model: Evidence from China-US

PONE-D-23-42358R1

Dear Dr. LEI,

We’re pleased to inform you that your manuscript has been judged scientifically suitable for publication and will be formally accepted for publication once it meets all outstanding technical requirements.

Kind regards,

Juan E. Trinidad Segovia, PhD

Section Editor

PLOS ONE

Additional Editor Comments (optional):

Reviewers' comments:

Reviewer's Responses to Questions

**Comments to the Author**

1. If the authors have adequately addressed your comments raised in a previous round of review and you feel that this manuscript is now acceptable for publication, you may indicate that here to bypass the “Comments to the Author” section, enter your conflict of interest statement in the “Confidential to Editor” section, and submit your "Accept" recommendation.

Reviewer #1: All comments have been addressed

Reviewer #2: All comments have been addressed

2. Is the manuscript technically sound, and do the data support the conclusions?

Reviewer #1: Yes

Reviewer #2: Yes

3. Has the statistical analysis been performed appropriately and rigorously? 

Reviewer #1: Yes

Reviewer #2: Yes

4. Have the authors made all data underlying the findings in their manuscript fully available?

Reviewer #1: Yes

Reviewer #2: Yes

5. Is the manuscript presented in an intelligible fashion and written in standard English?

Reviewer #1: Yes

Reviewer #2: Yes

6. Review Comments to the Author

Reviewer #1: Recommendation: Accept

Author carefully revised manuscript according to reviewers’ comments and suggestions. Response letter is written well to reply to questions proposed by reviewers one by on.

Reviewer #2: (No Response)

7. PLOS authors have the option to publish the peer review history of their article (what does this mean?). If published, this will include your full peer review and any attached files.

Reviewer #1: No

Reviewer #2: No
